# Coupled Gradient Estimators for Discrete Latent Variables

**Zhe Dong**
Google Research, Brain Team
zhedong@google.com

**Andriy Mnih**
DeepMind
andriy@deepmind.com

**George Tucker**
Google Research, Brain Team
gjt@google.com

## Abstract

Training models with discrete latent variables is challenging due to the high variance of unbiased gradient estimators. While low-variance reparameterization gradients of a continuous relaxation can provide an effective solution, a continuous relaxation is not always available or tractable. Dong et al. (2020) and Yin et al. (2020) introduced a performant estimator that does not rely on continuous relaxations; however, it is limited to binary random variables. We introduce a novel derivation of their estimator based on importance sampling and statistical couplings, which we extend to the categorical setting. Motivated by the construction of a stick-breaking coupling, we introduce gradient estimators based on reparameterizing categorical variables as sequences of binary variables and Rao-Blackwellization. In systematic experiments, we show that our proposed categorical gradient estimators provide state-of-the-art performance, whereas even with additional Rao-Blackwellization, previous estimators (Yin et al., 2019) underperform a simpler REINFORCE with a leave-one-out-baseline estimator (Kool et al., 2019).

## 1  Introduction

Optimizing an expectation of a cost function of discrete variables with respect to the parameters of their distribution is a frequently encountered problem in machine learning. This problem is challenging because the gradient of the objective, like the objective itself, is an expectation over an exponentially large space of joint configurations of the variables. As the number of variables increases, these expectations quickly become intractable and thus are typically approximated using Monte Carlo sampling, trading a reduction in computation time for variance in the estimates. When such stochastic gradient estimates are used for learning, their variance determines the largest learning rate that can be used without making training unstable. Thus finding estimators with lower variance leads directly to faster training by allowing higher learning rates. For example, the use of the reparameterization trick to yield low-variance gradient estimates has been essential to the success of variational autoencoders (Kingma and Welling, 2014; Rezende et al., 2014). However, this estimator, also known as the pathwise derivative estimator (Glasserman, 2013), can only be applied to continuous random variables.

For discrete random variables, there are two common strategies for stochastic gradient estimation. The first one involves replacing discrete variables with continuous ones that approximate them as closely as possible (Maddison et al., 2017; Jang et al., 2017) and training the resulting relaxed system with the reparameterization trick. However, as after training, the system is evaluated with discrete variables, this approach is not guaranteed to perform well and requires a careful choice of the continuous relaxation. Moreover, evaluating the cost function at the relaxed values instead of the discrete ones is not always desirable or even possible. The second strategy, involves using the REINFORCE estimator

---

Code and additional information: `https://sites.google.com/view/disarm-estimator`.

(Williams, 1992), also known as the score-function (Rubinstein and Shapiro, 1990) or likelihood-ratio (Glynn, 1990) estimator, which, having fewer requirements than the reparameterization trick, also works with discrete random variables. As the simplest versions of this estimator tend to exhibit high variance, they are typically combined with variance reduction techniques. Some of the most effective such estimators (Tucker et al., 2017; Grathwohl et al., 2018), incorporate the gradient information provided by the continuous relaxation, while keeping the estimator unbiased w.r.t. the original discrete system.

The recently introduced Augment-REINFORCE-Merge (ARM) (Yin and Zhou, 2019) estimator for binary variables and Augment-REINFORCE-Swap (ARS) and Augment-REINFORCE-Swap-Merge (ARSM) estimators (Yin et al., 2019) for categorical variables provide a promising alternative to relaxation-based estimators. However, when compared to a simpler baseline approach (REINFORCE with a leave-one-out-baseline (RLOO; Kool et al., 2019)), ARM underperforms in the binary setting (Dong et al., 2020), and we similarly demonstrate that ARS and ARSM underperform in the categorical setting. Dong et al. (2020) and Yin et al. (2020) independently developed an estimator that uses Rao-Blackwellization to improve ARM and outperforms RLOO, providing state-of-the-art performance in the binary setting.

In this paper, we explore how to devise a performant estimator in the categorical setting. A natural first approach is to apply the ideas from (Dong et al., 2020; Yin et al., 2020) to ARS and ARSM. However, empirically, we find that this is insufficient to close the gap between ARS/ARSM and RLOO. Due to space constraints, we defer the details to Appendix A.6. Instead, we develop a novel derivation of DisARM/U2G (Dong et al., 2020; Yin et al., 2020) using importance sampling which is simpler and more direct while providing a natural extension to the categorical case. This estimator requires constructing a *coupling* on categorical variables, which we do using a stick-breaking process and antithetic Bernoulli variables. Motivated by this construction, we also consider estimators based on reparameterizing the problem with a sequence of binary variables. We systematically evaluate these estimators and their underlying design choices and find that they outperform RLOO across a range of problems without requiring more computation.

## 2 Background

We consider the problem of optimizing

$$\mathbb{E}_{q_\theta(z)}\left[f_\theta(z)\right], \tag{1}$$

with respect to the parameters $\theta$ of a factorial categorical distribution $q_\theta(z) = \prod_k \text{Cat}(z_k; \alpha_{\theta,k})$ where $k$ indexes dimension and $\alpha_{\theta,k}$ is the vector of logits of the categorical distribution with $C$ choices.[1] This situation covers many problems with discrete latent variables, for example, in variational inference $f_\theta(z)$ could be the instantaneous ELBO (Jordan et al., 1999) and $q_\theta(z)$ the variational posterior.

The gradient with respect to $\theta$ is

$$\nabla_\theta \mathbb{E}_{q_\theta(z)}\left[f_\theta(z)\right] = \mathbb{E}_{q_\theta(z)}\left[f_\theta(z)\nabla_\theta \log q_\theta(z) + \nabla_\theta f_\theta(z)\right]. \tag{2}$$

It typically suffices to estimate the second term with a single Monte Carlo sample, so for notational clarity, we omit the dependence of $f$ on $\theta$ in the following sections. Monte Carlo estimates of the first term can have large variance. Low-variance, unbiased estimators of the first term will be our focus.

### 2.1 DisARM/U2G

Dong et al. (2020) and Yin et al. (2020) derived a Rao-Blackwellized estimator for *binary* variables based on the coupled estimator of Yin and Zhou (2019). In particular, if $q_\theta(b) \sim \prod_k \text{Bernoulli}(\alpha_{\theta,k})$ with $\theta$ parameterizing the logits $\alpha_{\theta_k}$ of the Bernoulli distribution, and $b$ and $\tilde{b}$ are antithetic samples[2] from $q_\theta(b)$ (independent across dimension $k$), then the estimator

$$\frac{1}{2}\left(f(b) - f(\tilde{b})\right)\left((-1)^{\tilde{b}_k}\mathbb{1}_{b_k \neq \tilde{b}_k}\sigma(|(\alpha_{\theta,k})|)\right)$$

---

[1] To simplify notation, we omit the subscripted $\theta$ on $\alpha$ when it is clear from context.

[2] Antithetic Bernoulli samples can be defined by the following process: sample $u \sim \text{Uniform}(0,1)$, then set $b = u < p$ and $\tilde{b} = (1-u) < p$, where $p$ is the probability parameter of the Bernoulli variable.

is an unbiased estimator of the gradient $\nabla_{\alpha_{\theta,k}} \mathbb{E}_{q_\theta(b)}[f(b)]$. This estimator has been shown to outperform RLOO, but is limited to the binary variable case.

## 3 Methods

First, we provide a novel derivation of DisARM (Dong et al., 2020) / U2G (Yin et al., 2020) from an importance-sampling perspective, which naturally extends to the categorical case. Starting with the (2-sample) REINFORCE LOO estimator (Kool et al., 2019) and applying importance sampling, we have

$$
\nabla_{\theta_k} \mathbb{E}_{q_\theta(z)}[f(z)] = \frac{1}{2} \mathbb{E}_{q_\theta(z)q_\theta(\tilde{z})} \left[ \overbrace{(f(z) - f(\tilde{z}))\left(\nabla_{\theta_k} \log q_\theta(z_k) - \nabla_{\theta_k} \log q_\theta(\tilde{z}_k)\right)}^{\text{RLOO}(z,\tilde{z})} \right]
$$

$$
= \frac{1}{2} \mathbb{E}_{p_\theta(z,\tilde{z})} \left[ \frac{q_\theta(z)q_\theta(\tilde{z})}{p_\theta(z,\tilde{z})}(f(z) - f(\tilde{z}))\left(\nabla_{\theta_k} \log q_\theta(z_k) - \nabla_{\theta_k} \log q_\theta(\tilde{z}_k)\right) \right].
$$

To reduce the variance of the estimator, we can use the joint distribution $p_\theta(z, \tilde{z})$ to emphasize terms that have high magnitude. However, controlling the weights can be challenging for high dimensional $z$. We can sidestep this issue by taking advantage of the structure of the integrand and requiring that $p_\theta$ be a properly supported[3] *coupling* that is independent across dimensions (i.e., a joint distribution $p(z, \tilde{z}) = \prod_k p(z_k, \tilde{z}_k)$ such that the marginals are maintained $p(z_k) = p(\tilde{z}_k) = q(z_k)$). Then with $(z, \tilde{z}) \sim p$, the following will be an unbiased estimator of the gradient (see Appendix A.3 for the derivation)

$$
g_{\text{DisARM-IW}\,k} = \frac{1}{2} \frac{q_\theta(z_k)q_\theta(\tilde{z}_k)}{p_\theta(z_k, \tilde{z}_k)}(f(z) - f(\tilde{z}))\left(\nabla_{\theta_k} \log q_\theta(z_k) - \nabla_{\theta_k} \log q_\theta(\tilde{z}_k)\right). \tag{3}
$$

Critically, because the coupling maintains the marginal distributions, we only need to importance weight a single dimension at a time, which ensures the weights are reasonable. The estimator is unbiased, so the coupling can be designed to reduce variance. In the binary case, with an antithetic Bernoulli coupling, it is straightforward to see that this precisely recovers DisARM/U2G. We note that Dimitriev and Zhou (2021) independently and concurrently discovered a similar result. Conveniently, this estimator is also valid in the categorical case, however, the choice of coupling affects the performance of the estimator.

**Stick-breaking Coupling**

The ideal variance-reducing coupling would take into account the magnitude of $(f(z) - f(\tilde{z}))\left(\nabla_{\theta_k} \log q_\theta(z_k) - \nabla_{\theta_k} \log q_\theta(\tilde{z}_k)\right)$ which we do not know a priori. However, we do know that when $z_k = \tilde{z}_k$, the last multiplicative term of Eq. 3 vanishes, so moving mass away from this configuration will reduce variance. Furthermore, for the estimator to be valid, the coupling must put non-zero mass on all $z_k \neq \tilde{z}_k$ configurations.[4]

In the binary case, there is a natural construction of an antithetic coupling which minimizes the probability of $z_k = \tilde{z}_k$. We can extend this construction using formulations of categorical variables as a function of a sequence of binary decisions. The stick-breaking construction for categorical variables (Khan et al., 2012) provides such an approach. Suppose we have a categorical distribution $q(z_k)$ and we construct a sequence of independent binary variables $b_{k,1}, \ldots, b_{k,C}$ with $b_{k,i} \sim \text{Bernoulli}(q(z_k = i)/\sum_{j=i}^{C} q(z_k = j))$ and define $z_k := \text{SB}(b_{k,1}, \ldots, b_{k,C}) := \min i$ s.t. $b_{k,i} = 1$; then $z_k \sim q(z_k)$. Given each of $b_{k,1}, \ldots, b_{k,C}$, we can independently sample antithetic Bernoulli variables $\tilde{b}_{k,1}, \ldots, \tilde{b}_{k,C}$ and let $\tilde{z} := \text{SB}(\tilde{b}_{k,1}, \ldots, \tilde{b}_{k,C})$. This process defines a joint distribution on $(z_k, \tilde{z}_k)$, which we call the *stick-breaking coupling*. By construction, the joint distribution preserves the marginal distributions. Note that the stick-breaking coupling depends on the an arbitrary ordering of the categories.

---

[3]As is standard for importance sampling, to ensure unbiasedness, we require that either $q(z, \tilde{z}) = 0$ or the integrand is 0 whenever $p(z, \tilde{z}) = 0$.

[4]Technically, the coupling can put zero mass on configurations as long as the expectation is zero across those configurations, however, this is hard to ensure for a general $f$.

Without reordering the categories, the coupling arising from the default ordering may not put non-zero mass on all $z_k \neq \tilde{z}_k$ configurations. When $p(b_{k,i}) \geq 0.5$, then $p(b_{k,i} = 0, \tilde{b}_{k,i} = 0) = 0$, which implies that $p(z_k > i, \tilde{z}_k > i) = 0$ by definition. So if $p(b_{k,i}) > 0.5$ for $i < C$, this violates the condition that the coupling put mass on all $z_k \neq \tilde{z}_k$ configurations. We can avoid this situation by relabeling the categories in the ascending order of probability (as this guarantees that $q(z_k = i)/\sum_{j=i}^{C} q(z_k = j) \leq 0.5$). With this ordering, computing the required importance weights with the stick-breaking coupling is straightforward when $z_k \neq \tilde{z}_k$

$$\frac{q_\theta(z_k)q_\theta(\tilde{z}_k)}{p_\theta(z_k, \tilde{z}_k)} = \prod_{i=1}^{\min(z_k, \tilde{z}_k)-1} \frac{\sigma(-\alpha_{k,i})^2}{1 - 2\sigma(\alpha_{k,i})} \sigma\left(-\alpha_{\min(z_k, \tilde{z}_k)}\right),$$

where $\alpha_{k,i} = \text{logit} \frac{q(z_k=i)}{\sum_{j=i}^{C} q(z_k=j)}$ (see Appendix A.4 for details). Note that due to the ascending ordering, the $\sigma(\alpha_{k,i})$ term in the product is always $\leq \frac{1}{3}$, ensuring that the weights are bounded.

Finally, we note that a mixture of couplings is also a coupling, so we can take any coupling and mix it with the independent coupling to ensure the support condition is met. Because the estimator is unbiased for any choice of nonzero mixing coefficient, the mixing coefficients can be optimized following the control variate tuning process described in (Ruiz et al., 2016; Tucker et al., 2017; Grathwohl et al., 2018). This also ensures that the performance of the estimator will be at least as strong as RLOO. Exploring this idea could be an interesting future direction.

### 3.1 Binary Reparameterization

Motivated by the stick-breaking construction, we can eschew importance sampling and directly reparameterize the problem in terms of the binary variables and apply DisARM/U2G. In particular, if we have binary variables $b_{k,1} \sim \text{Bernoulli}(\sigma(\alpha_{k,1})), \ldots, b_{k,C} \sim \text{Bernoulli}(\sigma(\alpha_{k,C}))$ such that $z_k := h(b_{k,1}, \ldots, b_{k,C}) \sim q_\theta(z_k)$ where $h$ is a deterministic function that does not depend on $\theta$, then we can rewrite the gradient as

$$\nabla_{\alpha_{k,c}} \mathbb{E}_{q_\theta(z)}[f(h(b))] = \nabla_{\alpha_{k,c}} \mathbb{E}_b[f(h(b))] = \mathbb{E}_b[f(h(b))\nabla_{\alpha_{k,c}} \log q_\theta(b)]$$
$$= \mathbb{E}_b[f(h(b))\nabla_{\alpha_{k,c}} \log q_\theta(b_{k,c})],$$

which is an expression that DisARM/U2G can be applied to.

We can additionally leverage the structure of the stick-breaking construction to further improve the estimator. Intuitively, by the definition of $z_k$ in terms of $\text{SB}(\cdots)$, if we know that $b_{k,c} = 1$, then the values of $b_{k,c'}$ for $c' > c$ do not change the value of $z_k$, which implies the gradient terms vanishing. In particular, given antithetically coupled Bernoulli samples, the following is an unbiased estimator (see Appendix A.4 for details):

$$g_{\text{DisARM-SB}_{k,c}} = \begin{cases} \frac{1}{2}\left(f(z) - f(\tilde{z})\right)\left((-1)^{\tilde{b}_{k,c}} \mathbb{1}_{b_{k,c} \neq \tilde{b}_{k,c}} \sigma(|(\alpha_{k,c}|)\right) & c \leq \min(z_k, \tilde{z}_k) \\ \frac{1}{2}\left(f(\tilde{z}) - f(z)\right)\nabla_{\alpha_{k,c}} \log q_\theta(\tilde{b}_{k,c}) & z_k < c \leq \tilde{z}_k \\ \frac{1}{2}\left(f(z) - f(\tilde{z})\right)\nabla_{\alpha_{k,c}} \log q_\theta(b_{k,c}) & \tilde{z}_k < c \leq z_k \\ 0 & c > \max(z_k, \tilde{z}_k) \end{cases} . \quad (4)$$

In contrast to the previous section, *any* choice of ordering for the stick-breaking construction results in an unbiased estimator, and we experimentally evaluated several choices.

Alternatively, we can use a sequence of independent Bernoulli variables arranged as a binary tree to construct categorical variables, with each leaf node representing a category (Ghahramani et al., 2010). For simplicity, we only consider balanced binary trees (i.e., $C$ a power of 2). The binary variables encode the internal routing decisions through the tree, defining a deterministic mapping from the binary variables to the categorical variable $z := T(b_1, \ldots, b_{C-1})$. It is straightforward to derive the Bernoulli probabilities from $q_\theta(z)$ so that $z \sim q_\theta(z)$, so we defer the details to Appendix A.5.

As above, we can additionally leverage the structure of the tree construction to further improve the estimator. Given binary variables $b_1, \ldots, b_{C-1}$, let $I(b_1, \ldots, b_{C-1})$ be the set of variables used in routing decisions ($|I(b_1, \ldots, b_{C-1})| = \log_2 C$). By the definition of $z_k$ in terms of $T(\ldots)$, we know that $z_k$ is unaffected by binary decisions that occur outside $I(b_{\cdot})$, which implies the gradient terms

vanishing. With a pair of antithetically sampled binary sequences, the following is an unbiased estimator (see Appendix A.5 for details):

$$g_{\text{DisARM-Tree}\,k,c} = \begin{cases} \frac{1}{2}\left(f(z) - f(\tilde{z})\right)\left((-1)^{\tilde{b}_{k,c}} \mathbb{1}_{b_{k,c} \neq \tilde{b}_{k,c}}\, \sigma(|(\alpha_{k,c}|)\right) & c \in I(b_{k,\cdot}) \cap I(\tilde{b}_{k,\cdot}) \\ \frac{1}{2}\left(f(\tilde{z}) - f(z)\right) \nabla_{\alpha_{k,c}} \log q_\theta(\tilde{b}_{k,c}) & c \in I(\tilde{b}_{k,\cdot}) - I(b_{k,\cdot}) \\ \frac{1}{2}\left(f(z) - f(\tilde{z})\right) \nabla_{\alpha_{k,c}} \log q_\theta(b_{k,c}) & c \in I(b_{k,\cdot}) - I(\tilde{b}_{k,\cdot}) \\ 0 & c \notin I(b_{k,\cdot}) \cup I(\tilde{b}_{k,\cdot}) \end{cases}. \quad (5)$$

Again, the choice of ordering affects the coupling, however, we leave investigating the optimal choice to future work.

Notably both estimators include terms that are reminiscent of the LOO estimator, however, they use *dependent* $z, \tilde{z}$ and are still unbiased due to careful construction.

## 3.2 Multi-sample extension

We can extend the proposed estimators to the case when we have $n$-coupled pairs $(z^1, \tilde{z}^1), \ldots, (z^n, \tilde{z}^n)$. Dimitriev and Zhou (2021) note that the $2n$-sample RLOO estimator can be written as an average over all 2-sample RLOO estimators. In other words, given $2n$ independent samples $z^1, \ldots, z^{2n}$,

$$g_{\text{RLOO}}(z^1, \ldots, z^{2n}) = \frac{1}{2n(2n-1)} \sum_{i \neq j} g_{\text{RLOO}}(z^i, z^j).$$

If instead of $2n$ independent samples, we used $n$-coupled pairs in the estimator, most 2-sample RLOO estimators will be with independent samples, however, $n$ estimators will use coupled pairs which introduces a bias. We form the unbiased $n$-coupled pair estimator by adding a correction term that replaces those terms with a DisARM estimator

$$g_{\text{RLOO}}(z^1, \tilde{z}^1, \ldots, z^n, \tilde{z}^n) + \frac{2}{2n(2n-1)} \sum_{i=1}^{n} (g_{\text{DisARM-*}}(z^i, \tilde{z}^i) - g_{\text{RLOO}}(z^i, \tilde{z}^i)),$$

where we can use any of the previous DisARM-* estimators.

## 4 Related Work

Almost all unbiased gradient estimators for discrete random variables (RVs), including our proposed estimators, belong to the REINFORCE / score function / likelihood-ratio estimator family (Williams, 1992; Rubinstein and Shapiro, 1990; Glynn, 1990). The primary difference between such estimators is the variance reduction techniques they incorporate, trading additional computation for lower variance of the estimate. Averaging over multiple independent samples is the simplest but relatively inefficient variance reduction technique, as the variance of the estimate is inversely proportional to the number of samples and thus decreases slowly. REINFORCE with the leave-out-out baseline (RLOO; Kool et al., 2019) provides a much more effective way of utilizing multiple independent samples, by using the average over all the other samples as the baseline for each sample. Kool et al. (2020) extend this approach by sampling *without* replacement, however, they find that in the high-dimensional spaces we consider, we almost never see such duplicates, resulting in little to no improvement in sampling without replacement compared to sampling with replacement (Figures 2(b) and 4(b) in (Kool et al., 2020)).

The estimators we develop use a technique called *coupling* (Owen, 2013) to make better use of multiple samples by making them dependent. Recently, this kind of approach has been applied to binary RVs, resulting in the Augment-REINFORCE-Merge (ARM; Yin and Zhou, 2019) and DisARM (Dong et al., 2020) / U2G (Yin et al., 2020) estimators. ARM is constructed by reparameterizing Bernoulli RVs in terms of Logistic variables and applying antithetic sampling, which is the simplest kind of coupling, to the resulting REINFORCE estimator. DisARM / U2G is obtained by Rao-Blackwellizing the ARM estimator to eliminate its dependence on the values of the underlying Logistic variables, reducing its variance. We show how to employ DisARM with categorical RVs by representing them as sequences of binary decisions and applying DisARM to the resulting system. By considering the sequences obtained through stick-breaking and tree-structured decisions, we obtain

Table 1: Mean variational lower bounds and the standard error of the mean computed based on 5 runs of $5 \times 10^5$-steps training from different random initializations. The best performing method (up to the standard error) for each task is in bold. Top: comparing the performance of the proposed estimators and the RLOO baseline on three datasets. Bottom: comparing the effect of arranging the categories in the ascending or descending order to the default ordering for DisARM-SB.

| | Estimator Comparison with Train ELBO | | | | |
|---|---|---|---|---|---|
| | DisARM-IW | DisARM-Tree | DisARM-SB | RLOO | RELAX |
| DynamicMNIST | $-103.41 \pm 0.23$ | $\mathbf{-103.10 \pm 0.25}$ | $-103.35 \pm 0.17$ | $-104.03 \pm 0.23$ | $-102.22 \pm 0.34$ |
| FashionMNIST | $-240.08 \pm 0.17$ | $\mathbf{-239.83 \pm 0.29}$ | $\mathbf{-239.66 \pm 0.20}$ | $-240.89 \pm 0.49$ | $-238.81 \pm 0.34$ |
| Omniglot | $\mathbf{-122.25 \pm 0.35}$ | $-123.08 \pm 0.43$ | $\mathbf{-121.99 \pm 0.17}$ | $\mathbf{-122.70 \pm 0.80}$ | $-120.40 \pm 0.16$ |

| | Ordering Comparison with Train ELBO | | |
|---|---|---|---|
| | Ascending | Descending | Default |
| DynamicMNIST | $\mathbf{-103.35 \pm 0.17}$ | $\mathbf{-103.56 \pm 0.28}$ | $\mathbf{-103.46 \pm 0.17}$ |
| FashionMNIST | $\mathbf{-239.66 \pm 0.20}$ | $-240.04 \pm 0.24$ | $-239.99 \pm 0.20$ |
| Omniglot | $\mathbf{-121.99 \pm 0.17}$ | $-122.79 \pm 0.12$ | $-122.75 \pm 0.15$ |

the DisARM-SB and DisARM-Tree estimators respectively. We obtain DisARM-IW by generalizing the two-sample version of RLOO to allow the two samples to be coupled and using an importance-weighting correction to keep the estimator unbiased. DisARM-IW reduces to DisARM in the case of Bernoulli RVs and antithetic coupling, and thus can be seen as a generalization. Dimitriev and Zhou (2021) has concurrently and independently developed an estimator identical to DisARM-IW, though they considered only the case of Bernoulli random variables.

Augment-REINFORCE-Swap estimator (ARS; Yin et al., 2019) is a generalization of ARM to categorical RVs; it reparameterizes the categorical variables in terms of Dirichlet RVs and applies REINFORCE to them, using a coupling based on a swapping operation. Augment-REINFORCE-Swap-Merge (ARSM; Yin et al., 2019) goes one step further by averaging over all possible choices of the pivot dimension that ARS chooses arbitrarily. Our Rao-Blackwellized ARS+ and ARSM+ estimators are obtained by analytically integrating out the randomness introduced by the augmenting Dirichlet dimensions from the ARS and ARSM estimators respectively, following the strategy that was used to derive DisARM from ARM (see Appendix A.6 for details).

Rao-Blackwellization has been a popular variance reduction technique in the literature on gradient estimation with discrete random variables, having been used in BBVI (Ranganath et al., 2014), NVIL (Mnih and Gregor, 2014), LEG (Titsias and Lázaro-Gredilla, 2015), as well as more recently by Liu et al. (2019) and Paulus et al. (2020). Stick-breaking has previously been used in the context of VAEs to allow inferring the latent space dimensionality (Nalisnick and Smyth, 2016).

In this paper we use variational training of latent variable models as a benchmark for unbiased gradient estimators for general expectations rather than a goal in itself. As a result, we do not compare to specialized techniques for training models with discrete latent variables, such as Reweighted Wake Sleep (Bornschein and Bengio, 2015) and Joint Stochastic Approximation (Ou and Song, 2020), which do not provide such unbiased estimators and optimize different objectives.

## 5 Experiments

Our goal was to derive unbiased, low-variance gradient estimators for the categorical setting, which do not rely on continuous relaxation. We primarily compared against the REINFORCE estimator with a leave-one-out baseline (RLOO, Kool et al., 2019). In the high dimensional setting that we evaluated, RLOO has been shown to be a strong baseline algorithm, performing at least as well as more complex methods, such as RELAX, REBAR, STORB, and UnOrd, as shown in (Richter et al., 2020; Kool et al., 2020). As in prior work (Yin et al., 2019), we benchmark the estimators by training variational auto-encoders (Kingma and Welling, 2014; Rezende et al., 2014) (VAE) with *categorical* latent variables on three datasets: binarized MNIST[5] (LeCun et al., 2010), FashionMNIST[6] (Xiao

---

[5]Creative Commons Attribution-Share Alike 3.0 license
[6]MIT license

Table 2: Evaluating the estimators and the effect of ordering on FashionMNIST with different model sizes. Mean variational lower bounds and the standard error of the mean computed based on 5 runs of $5 \times 10^5$-steps training from different random initializations. We consider three combinations of the number of categories (C) and the number of latent variables (V). The best performing method (up to the standard error) for each task is in bold. Top: evaluating the performance of the proposed estimators and the RLOO baseline. Bottom: comparing the effect of arranging the categories in the ascending or descending order to the default ordering for DisARM-SB.

| Estimator Comparison with Train ELBO | | | | |
|---|---|---|---|---|
| | DisARM-IW | DisARM-Tree | DisARM-SB | RLOO |
| C64 / V5 | $\mathbf{-244.75 \pm 0.53}$ | $\mathbf{-244.29 \pm 0.30}$ | $\mathbf{-244.81 \pm 0.64}$ | $-245.27 \pm 0.30$ |
| C64 / V32 | $-240.08 \pm 0.17$ | $\mathbf{-239.83 \pm 0.29}$ | $\mathbf{-239.66 \pm 0.20}$ | $-240.89 \pm 0.49$ |
| C16 / V32 | $\mathbf{-239.94 \pm 0.14}$ | $-240.52 \pm 0.22$ | $-240.43 \pm 0.13$ | $-240.99 \pm 0.25$ |

| Ordering Comparison with Train ELBO | | |
|---|---|---|
| | Ascending | Descending | Default |
| C64 / V5 | $\mathbf{-244.81 \pm 0.64}$ | $\mathbf{-244.95 \pm 0.36}$ | $\mathbf{-244.61 \pm 0.38}$ |
| C64 / V32 | $\mathbf{-239.66 \pm 0.20}$ | $-240.04 \pm 0.24$ | $-239.99 \pm 0.20$ |
| C16 / V32 | $\mathbf{-240.43 \pm 0.13}$ | $\mathbf{-240.39 \pm 0.21}$ | $\mathbf{-240.50 \pm 0.13}$ |

et al., 2017), and Omniglot[7] (Lake et al., 2015). As we sought to evaluate optimization performance, we use dynamic binarization to avoid overfitting and largely find that training performance mirrors test performance. We use the standard split into train, validation, and test sets.

To allow direct comparison, we use the same model structure as in (Yin et al., 2019). Briefly, the model has a single layer of categorical latent variables which are then represented as one-hot vectors and decoded to Bernoulli logits for the observation through either a linear transformation or an MLP with two hidden layers of 256 and 512 of LeakyReLU units (Xu et al., 2015). The encoder mirrors the structure, with two hidden layers of 512 and 256 LeakyReLU units and outputs producing the softmax logits. For most experiments, we used 32 latent variables with 64 categories unless specified otherwise. See Appendix A.2 for more details.

## 5.1 Evaluating DisARM-IW, DisARM-SB & DisARM-Tree

First, we evaluated the importance-weighted estimator (DisARM-IW), and DisARM estimators with the stick breaking construction (DisARM-SB) and the tree structured construction (DisARM-Tree). These estimators require only 2 expensive function evaluations regardless of the number of categories ($C$), unlike ARS/ARSM which require up to $O(C)/O(C^2)$ evaluations, so we use the 2-independent sample RLOO estimator as a baseline.

In the case of nonlinear models (Figure 1), DisARM-IW and DisARM-SB perform similarly, with lower gradient variance and better performance than the baseline estimator (RLOO) across all datasets. While for DisARM-IW, we have to use the ascending order for the stick-breaking construction to ensure unbiasedness, for DisARM-SB any ordering results in an unbiased estimator. We experimented with ordering the categories in the ascending and descending order by probability as well as using the default ordering (Figure 2). We found that the estimator with the ascending order consistently outperforms the other ordering schemes, though the gain is dataset-dependent. However, when we varied the number of categories and variables (Figure 6), we found that no ordering dominates. Given that ordering introduces an additional complexity, we recommend no ordering in practice.

In the case of linear models, we found that all estimators performed similarly (Appendix A.1). Dong et al. (2020) found that for multi-layer linear models in the binary case, DisARM showed increasing improvement over RLOO for models with deeper hierarchies. As nonlinear models are more common and expressive, we leave exploring whether this trend holds in the categorical case to future work.

We further verified that the observed improvements of the proposed estimators w.r.t. the RLOO baseline are consistent for models with different sizes of the latent space. As shown in Appendix

---

[7]MIT license

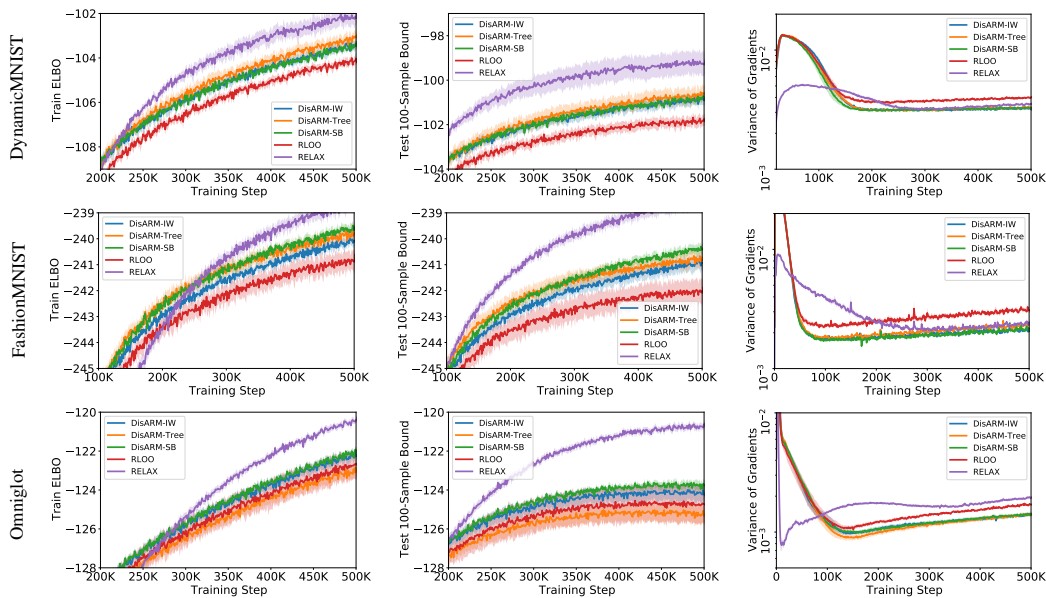

Figure 1: Training a non-linear categorical VAE with 32 latent variables with 64 categories on dynamically binarized MNIST, FashionMNIST, Omniglot datasets by maximizing the ELBO. We plot the train ELBO (left column), test 100-sample bound (middle column), and the variance of gradient estimator (right column). For a fair comparison, the variance of all the gradient estimators was computed along the training trajectory of the RLOO estimator. We plot the mean and one standard error based on 5 runs from different random initializations.

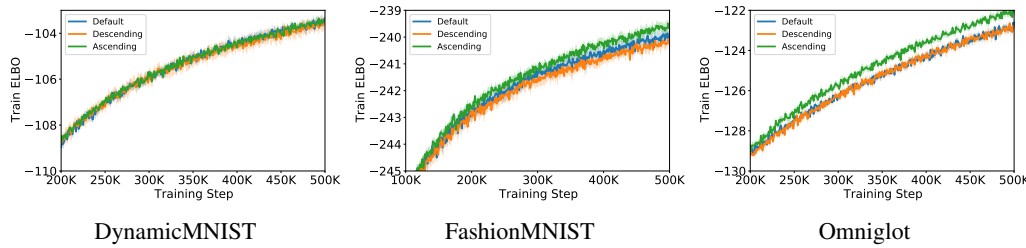

| DynamicMNIST | FashionMNIST | Omniglot |

Figure 2: The effect of logit ordering on the performance of DisARM-SB. We sort the encoder logits in the ascending or descending order and compare against the default ordering.

Figure 6, we find that the best choice of the proposed estimators depends on the model size, however all three estimators outperform RLOO in all cases.

### 5.2 Evaluating Multi-sample Estimators

We evaluate the multi-sample extension of the DisARM-* estimators on three benchmark datasets, by training a non-linear categorical VAE. The VAE has a stochastic hidden layer with 128 categorical latent variables, each with 16 categories. We run experiments with 5 antithetic pairs (10 samples) and 10 antithetic pairs (20 samples), and found that the proposed estimators outperform RLOO with a comparable number of independent samples. Even with the increasing number of samples, the performance improvement of the proposed estimators w.r.t. RLOO still holds, as shown in Figure 3, Appendix Figure 7, and Appendix Table 3.

## 6   Conclusion

We introduce a novel derivation of DisARM/U2G estimator based on importance sampling and statistical couplings, and naturally extend it to the categorical setting, calling the resulting estimator

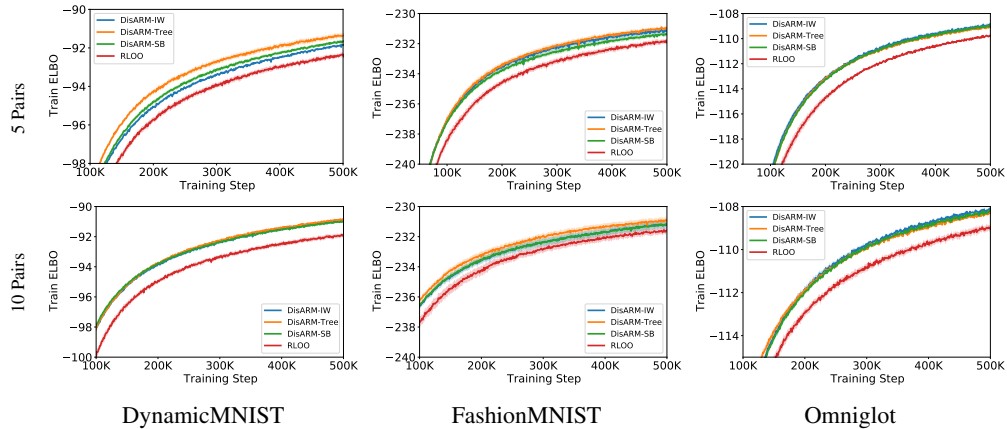

Figure 3: Training a non-linear categorical VAE with 128 latent variables and 16 categories, using multi-sample estimators, on dynamically binarized MNIST, FashionMNIST, Omniglot datasets by maximizing the ELBO. The examples in the top row are using 5 pairs of samples, while the ones in the bottom row are using 10 pairs.

DisARM-IW. Motivated by the construction of a stick-breaking coupling, we introduce two estimators, DisARM-SB and DisARM-Tree, by reparameterizing the problem with a sequence of binary variables and performing Rao-Blackwellization.

With systematic experiments, we demonstrate that the proposed estimators provide state-of-the-art performance. We find that the proposed estimators usually perform similarly and all outperform RLOO, with the winner depending on the dataset and the model. As the importance sampling estimator DisARM-IW is simpler to implement, more natural to understand, and easier to generalize, we recommend it in practice. We expect that this estimator can be further improved through better couplings, which is something we intend to explore in the future.

A limitation of the introduced categorical couplings is that they impose structure on the categorical space (i.e., an ordering or tree structure), which is not fully satisfactory because in most settings there is no such natural structure for categorical spaces. Developing coupling-based estimators that do not rely on such a structure would be interesting future work which might lead to further improvements. While we see that our coupling-based estimators generally outperform or perform at least as well as RLOO in our experiments, using coupled samples instead of independent samples is not guaranteed to lead to better performance. Learning the couplings would provide a way of ensuring an improvement over RLOO. Finally, the estimators we propose in this paper, like all multi-sample estimators, can be used for RL only if the environment is simulated or we have a model of it, as they require being able to perform multiple rollouts from the same state.

Discrete latent variables have particular applicability to interpretable models and sparse/conditional computation. Improving the foundational tools to train such models will make them more widely available. While interpretable systems are typically viewed as a positive, they only give a partial view of a complex system, and they can be misused to give a false sense of understanding. While sparse/conditional models can reduce the environmental cost of learned models (Patterson et al., 2021), the unintended implications of large models must also be considered (Bender et al., 2021).

## Acknowledgments and Disclosure of Funding

We thank Chris J. Maddison and Ben Poole for helpful comments. The authors did not receive any third-party funding or third-party support for this work, and have no financial relationships with entities that could potentially be perceived to influence what they wrote in the submitted work.

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
