# A Appendix

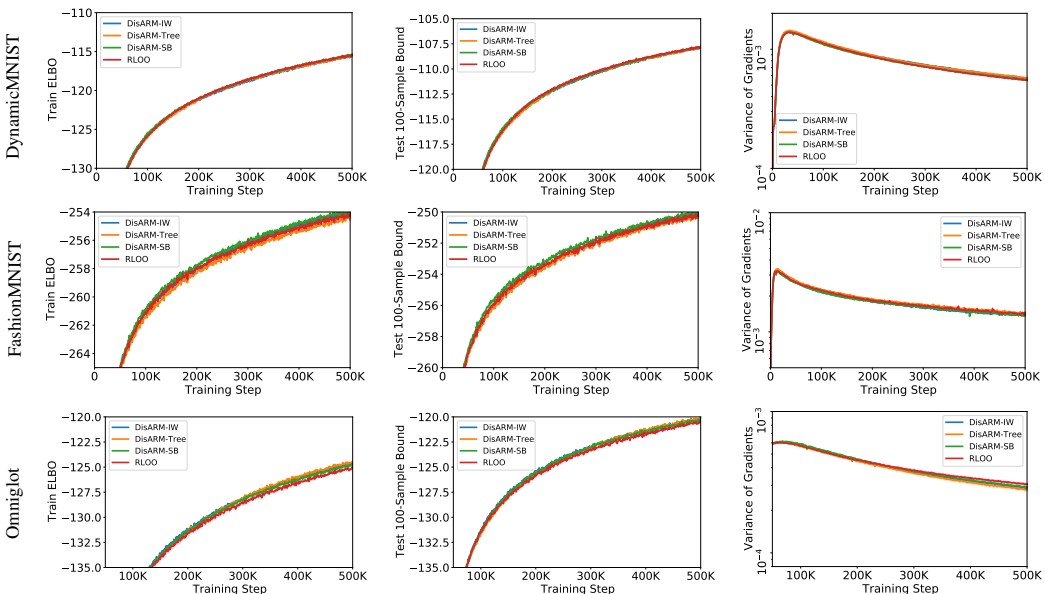

Figure 4: Training a linear VAE with 32 latent variables with 64 categories on dynamically binarized MNIST, FashionMNIST, and Omniglot datasets by maximizing the ELBO. We plot the train ELBO (left column), the test 100-sample bound (middle column), and the variance of the gradient estimator (right column). For a fair comparison, the variance of all the gradient estimators was computed along the training trajectory of the RLOO estimator. We plot the mean and one standard error based on 5 runs from different random initializations.

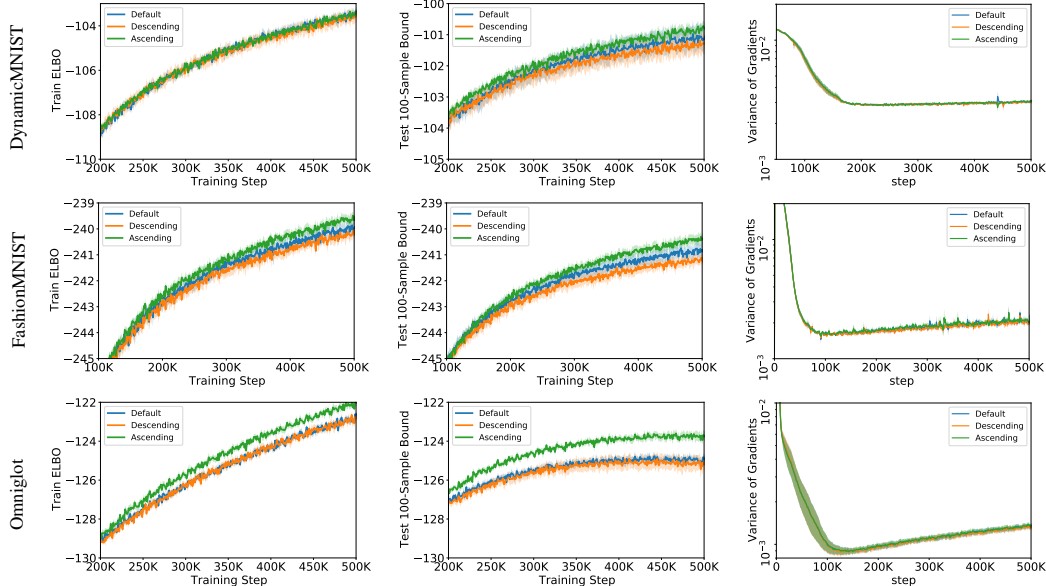

Figure 5: The effect of logit ordering on the performance of DisARM-SB. We sort the encoder logits in the ascending (Green) or descending (Red) order, and compare against the default ordering (Blue).

## A.1 Experiments with linear categorical VAEs

We evaluate the three proposed gradient estimators, DisARM-IW, DisARM-SB, and DisARM-Tree, by training linear variational auto-encoders with categorical latent variables on dynamically bina-

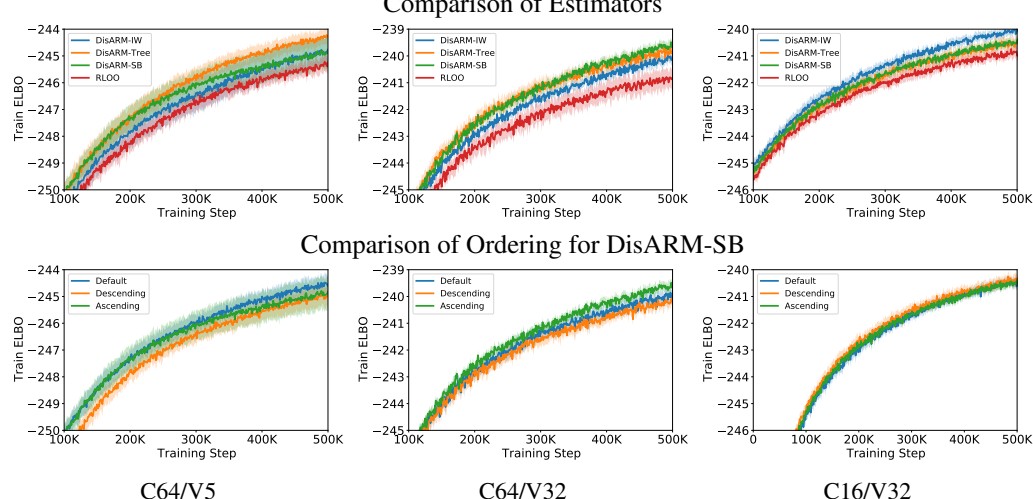

Figure 6: Training non-linear categorical VAEs with different model sizes on dynamically binarized FashionMNIST. Left: 5 latent variables of 64 categories. Middle: with 32 latent variables with 64 categories. Right: 32 latent variables with 16 categories.

Table 3: Training a non-linear VAE with categorical latents using multi-sample estimators. Mean variational lower bounds and the standard error of the mean computed based on 5 runs of $5 \times 10^5$-steps training from different random initializations. The best performing method (up to the standard error) for each task is in bold.

| #Pairs | Dataset | DisARM-Tree | DisARM-SB | DisARM-IW | RLOO |
|---|---|---|---|---|---|
| 5 | Dynamic MNIST | $-\mathbf{91.36 \pm 0.11}$ | $-91.67 \pm 0.08$ | $-91.85 \pm 0.04$ | $-92.35 \pm 0.10$ |
| | Fashion MNIST | $-\mathbf{231.01 \pm 0.14}$ | $-231.38 \pm 0.14$ | $-\mathbf{231.14 \pm 0.17}$ | $-231.78 \pm 0.16$ |
| | Omniglot | $-\mathbf{109.00 \pm 0.10}$ | $-\mathbf{108.90 \pm 0.09}$ | $-\mathbf{108.83 \pm 0.11}$ | $-109.73 \pm 0.07$ |
| 10 | Dynamic MNIST | $-\mathbf{90.81 \pm 0.06}$ | $-91.01 \pm 0.06$ | $-\mathbf{90.95 \pm 0.08}$ | $-91.86 \pm 0.15$ |
| | Fashion MNIST | $-\mathbf{230.95 \pm 0.21}$ | $-231.21 \pm 0.18$ | $-231.25 \pm 0.29$ | $-231.59 \pm 0.21$ |
| | Omniglot | $-108.27 \pm 0.03$ | $-108.19 \pm 0.07$ | $-\mathbf{108.08 \pm 0.06}$ | $-108.95 \pm 0.17$ |

rized MNIST, FashionMNIST, and Omniglot datasets. As in (Dong et al., 2020), we benchmark the proposed estimators against the 2-sample REINFORCE estimator with the leave-one-out baseline (RLOO; Kool et al., 2019). The linear model has a single layer of 32 categorical latent variables, each with 64 categories. We find no significant difference in performance between the proposed estimators and the RLOO baseline in this setting (Appendix Table 4 and Appendix Figure 4). However, as we noted in the maintext, Dong et al. (2020) found that for multi-layer linear models in the binary case, DisARM showed increasing improvement over RLOO for models with deeper hierarchies. So it would be interesting to see whether this holds for the categorical case in future work.

Table 4: Training a linear VAE with categorical latents using the proposed estimators and the RLOO baseline. Mean variational lower bounds and the standard error of the mean computed based on 5 runs of $5 \times 10^5$-steps training from different random initializations. The best performing method (up to the standard error) for each task is in bold.

| | Training set ELBO | | | |
|---|---|---|---|---|
| | DisARM-IW | DisARM-Tree | DisARM-SB | RLOO |
| Dynamic MNIST | $-115.64 \pm 0.09$ | $-\mathbf{115.59 \pm 0.16}$ | $-115.48 \pm 0.14$ | $-\mathbf{115.50 \pm 0.18}$ |
| Fashion MNIST | $-\mathbf{254.23 \pm 0.19}$ | $-254.56 \pm 0.22$ | $-\mathbf{254.05 \pm 0.20}$ | $-\mathbf{254.18 \pm 0.15}$ |
| Omniglot | $-124.64 \pm 0.02$ | $-\mathbf{124.52 \pm 0.11}$ | $-124.82 \pm 0.21$ | $-125.10 \pm 0.11$ |

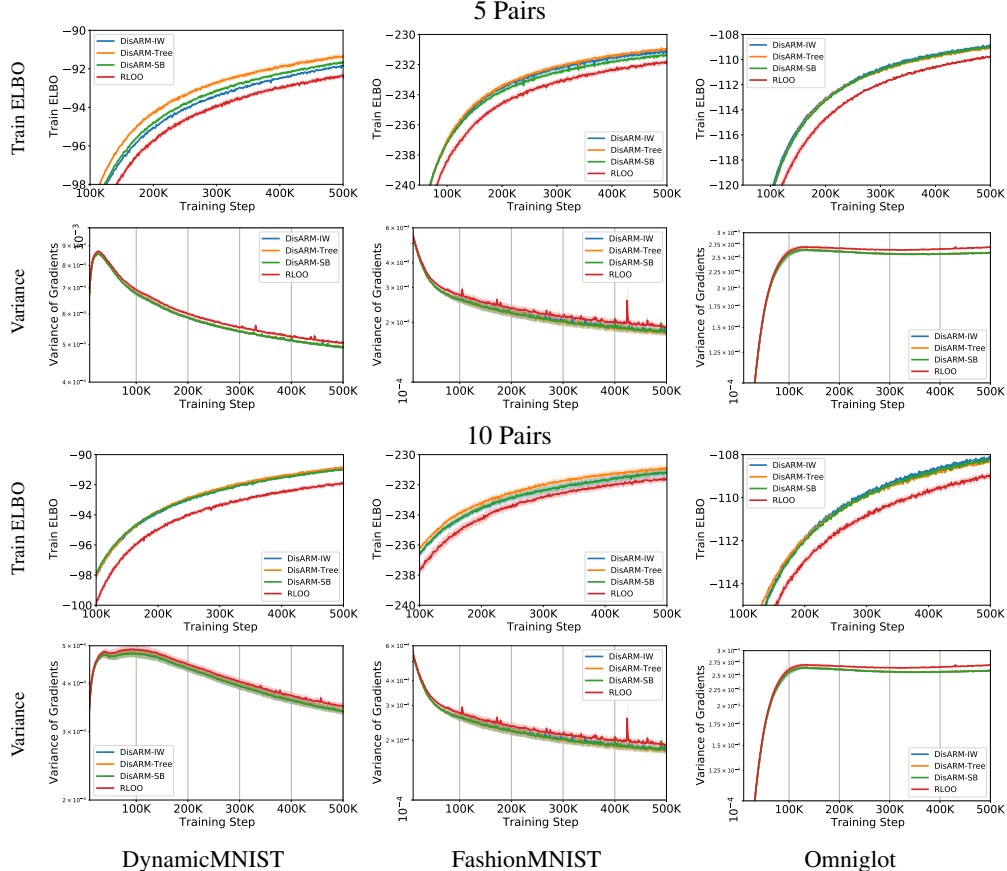

Figure 7: Training a non-linear categorical VAE with $128$ latent variables and $16$ categories, using multi-sample objectives, on dynamically binarized MNIST, FashionMNIST, Omniglot datasets by maximizing the ELBO. The examples in the top 2 rows are using 5 pairs of samples, while the ones in the bottom 2 rows are using 10 pairs.

## A.2 Experimental Details

We use the same model structure as in (Yin et al., 2019). The model has a single layer of categorical latent variables which are mapped to Bernoulli logits using an MLP with two hidden layers of $256$ and $512$ of LeakyReLU units (Xu et al., 2015) with negative slope of $0.2$. The encoder mirrors the structure, having two hidden layers of $512$ and $256$ LeakyReLU units.

For a fair comparison of the variance of the gradient estimators, we train a model with the RLOO estimator and evaluate the variance of all the estimators at each step along the training trajectory of this model. Based on preliminary experiments, the results were independent of the gradient estimator used to generate the model trajectory. We report the average per-parameter variance based on the parameter moments estimated with an exponential moving average with decay rate 0.999.

Each experiment run takes around $12$ hours on an NVIDIA Tesla P100 GPU. Our implementation was biased towards readability instead of computational efficiency, so we expect that significant improvements in runtime could be achieved.

## A.3 Importance Weighting Derivation

We consider proposal distributions that factorize across dimensions $p(z, \tilde{z}) = \prod_k p(z_k, \tilde{z}_k)$ and that are couplings such that the marginals are maintained $p(z_k) = p(\tilde{z}_k) = q(z_k)$. In general, $q_\theta(z) q_\theta(\tilde{z})$ has full support, but we know that the integrand $q_\theta(z) q_\theta(\tilde{z})(f(z) - f(\tilde{z})) (\nabla_{\theta_k} \log q_\theta(z_k) - \nabla_{\theta_k} \log q_\theta(\tilde{z}_k))$ vanishes when $z_k = \tilde{z}_k$ by inspection, so we allow $p(z_k, \tilde{z}_k)$ to put zero mass on $z_k = \tilde{z}_k$ configurations, and require $p(z_k, \tilde{z}_k) > 0$ otherwise.

Then with $z, \tilde{z} \sim p$, we show that $g_{\text{DisARM-IW}}$ (Eq. 3) is an unbiased estimator. First,

$$\mathbb{E}_{p(z,\tilde{z})} \left[ \frac{1}{2} \frac{q_\theta(z_k) q_\theta(\tilde{z}_k)}{p_\theta(z_k, \tilde{z}_k)} (f(z) - f(\tilde{z})) \left( \nabla_{\theta_k} \log q_\theta(z_k) - \nabla_{\theta_k} \log q_\theta(\tilde{z}_k) \right) \right]$$

$$= \mathbb{E}_{p(z_{-k}, \tilde{z}_{-k})} \mathbb{E}_{p(z_k, \tilde{z}_k)} \left[ \frac{1}{2} \frac{q_\theta(z_k) q_\theta(\tilde{z}_k)}{p_\theta(z_k, \tilde{z}_k)} (f(z) - f(\tilde{z})) \left( \nabla_{\theta_k} \log q_\theta(z_k) - \nabla_{\theta_k} \log q_\theta(\tilde{z}_k) \right) \right]$$

$$= \mathbb{E}_{p(z_{-k}, \tilde{z}_{-k})} \sum_{z_k, \tilde{z}_k \in \text{supp } p(z_k, \tilde{z}_k)} \frac{1}{2} q_\theta(z_k) q_\theta(\tilde{z}_k) (f(z) - f(\tilde{z})) \left( \nabla_{\theta_k} \log q_\theta(z_k) - \nabla_{\theta_k} \log q_\theta(\tilde{z}_k) \right)$$

$$= \mathbb{E}_{p(z_{-k}, \tilde{z}_{-k})} \sum_{z_k, \tilde{z}_k} \frac{1}{2} q_\theta(z_k) q_\theta(\tilde{z}_k) (f(z) - f(\tilde{z})) \left( \nabla_{\theta_k} \log q_\theta(z_k) - \nabla_{\theta_k} \log q_\theta(\tilde{z}_k) \right)$$

$$= \mathbb{E}_{p(z_{-k}, \tilde{z}_{-k})} \mathbb{E}_{q_\theta(z_k) q_\theta(\tilde{z}_k)} \left[ \frac{1}{2} (f(z) - f(\tilde{z})) \left( \nabla_{\theta_k} \log q_\theta(z_k) - \nabla_{\theta_k} \log q_\theta(\tilde{z}_k) \right) \right],$$

using the fact that the integrand vanishes outside the support of $p(z_k, \tilde{z}_k)$. Then,

$$\mathbb{E}_{p(z_{-k}, \tilde{z}_{-k})} \mathbb{E}_{q_\theta(z_k) q_\theta(\tilde{z}_k)} \left[ \frac{1}{2} (f(z) - f(\tilde{z})) \left( \nabla_{\theta_k} \log q_\theta(z_k) - \nabla_{\theta_k} \log q_\theta(\tilde{z}_k) \right) \right]$$

$$= \mathbb{E}_{q_\theta(z_k) q_\theta(\tilde{z}_k)} \left[ \frac{1}{2} (\mathbb{E}_{p(z_{-k}, \tilde{z}_{-k})} [f(z) - f(\tilde{z})]) \left( \nabla_{\theta_k} \log q_\theta(z_k) - \nabla_{\theta_k} \log q_\theta(\tilde{z}_k) \right) \right]$$

$$= \mathbb{E}_{q(z_k) q(\tilde{z}_k)} \left[ \frac{1}{2} (\mathbb{E}_{p(z_{-k}, \tilde{z}_{-k})} [f(z)] - \mathbb{E}_{p(z_{-k}, \tilde{z}_{-k})} [f(\tilde{z})]) \left( \nabla_{\theta_k} \log q_\theta(z_k) - \nabla_{\theta_k} \log q_\theta(\tilde{z}_k) \right) \right]$$

$$= \mathbb{E}_{q(z_k) q(\tilde{z}_k)} \left[ \frac{1}{2} (\mathbb{E}_{q(z_{-k})} [f(z)] - \mathbb{E}_{q(\tilde{z}_{-k})} [f(\tilde{z})]) \left( \nabla_{\theta_k} \log q_\theta(z_k) - \nabla_{\theta_k} \log q_\theta(\tilde{z}_k) \right) \right]$$

$$= \mathbb{E}_{q(z) q(\tilde{z})} \left[ \frac{1}{2} (f(z) - f(\tilde{z})) \left( \nabla_{\theta_k} \log q_\theta(z_k) - \nabla_{\theta_k} \log q_\theta(\tilde{z}_k) \right) \right],$$

following from the linearity of the expectation and the fact that the coupling preserves marginals.

### A.4  Stick-Breaking Coupling

#### A.4.1  Computing Importance Weights

To simplify the notation in this subsection, we consider a single dimension at a time and omit the dimension index. To compute $\frac{q_\theta(z) q_\theta(\tilde{z})}{p_\theta(z, \tilde{z})}$, it is helpful to work with the logits of the binary variables $\alpha_i = \text{logit} \frac{q(z=i)}{\sum_{j=i+1}^{C} q(z=j)}$. First, we know that $q(z=i) = \prod_{j=1}^{i-1} \sigma(-\alpha_j) \sigma(\alpha_i)$. For a pair of antithetic binary variables $(b_i, \tilde{b}_i)$, the coupling joint probability is

$$p(b_i, \tilde{b}_i) = \begin{cases} \max(1 - 2\sigma(\alpha_i), 0) & b_i = \tilde{b}_i = 0 \\ \max(2\sigma(\alpha_i) - 1, 0) & b_i = \tilde{b}_i = 1 \\ \sigma(-|\alpha_i|) & \text{o.w.} \end{cases}.$$

Since the categories are arranged in the ascending order of probability, $\alpha_i \leq 0$ for $i < C$, so the joint probability simplifies to

$$p(b_i, \tilde{b}_i) = \begin{cases} 1 - 2\sigma(\alpha_i) & b_i = \tilde{b}_i = 0 \\ 0 & b_i = \tilde{b}_i = 1 \\ \sigma(\alpha_i) & \text{o.w.} \end{cases},$$

for $i < C$. We do not need to compute the entries for $z = \tilde{z}$ as the integrand already vanishes. Because of symmetry, without loss of generality, assume $z < \tilde{z}$. From the construction of $z$ and $\tilde{z}$ in terms of the binary variables, we can reason about their values. We know that for $i < z$, we must have $b_i = \tilde{b}_i = 0$. Then, for $i = z$, we must have $b_i = 1$ and for $z \leq i < \tilde{z}$, $\tilde{b}_i = 0$, and finally, for $i = \tilde{z}, \tilde{b} = 1$. Putting this together yields

$$p(z, \tilde{z}) = \left[ \prod_{i=1}^{z-1} (1 - 2\sigma(\alpha_i)) \right] \sigma(\alpha_z) \left[ \prod_{j=z+1}^{\tilde{z}-1} \sigma(-\alpha_j) \right] \sigma(\alpha_{\tilde{z}}).$$

Thus, the importance weights are

$$\frac{q_\theta(z)q_\theta(\tilde{z})}{p_\theta(z,\tilde{z})} = \frac{\left[\prod_{i=1}^{z-1}\sigma(-\alpha_i)\right]\sigma(\alpha_z)\left[\prod_{j=1}^{\tilde{z}-1}\sigma(-\alpha_j)\right]\sigma(\alpha_{\tilde{z}})}{\left[\prod_{i=1}^{z-1}(1-2\sigma(\alpha_i))\right]\sigma(\alpha_z)\left[\prod_{j=z+1}^{\tilde{z}-1}\sigma(-\alpha_j)\right]\sigma(\alpha_{\tilde{z}})} = \frac{\left[\prod_{i=1}^{z-1}\sigma(-\alpha_i)^2\right]\sigma(-\alpha_z)}{\prod_{i=1}^{z-1}(1-2\sigma(\alpha_i))}.$$

### A.4.2 Unbiasedness of $g_{\text{DisARM-SB}}$

Recall the $g_{\text{DisARM-SB}}$ estimator. We have binary variables $b_{k,1} \sim \text{Bernoulli}(\sigma(\alpha_{k,1})), \ldots, b_{k,C} \sim$ $\text{Bernoulli}(\sigma(\alpha_{k,C}))$ and independently sampled antithetic pairs $\{\tilde{b}_{k,c}\}$ such that $z = h(b)$ and $\tilde{z} = h(\tilde{b})$. The estimator is

$$g_{\text{DisARM-SB}\,k,c} = \begin{cases} \frac{1}{2}\left(f(z)-f(\tilde{z})\right)\left((-1)^{\tilde{b}_{k,c}}\mathbb{1}_{b_{k,c}\neq\tilde{b}_{k,c}}\sigma(|(\alpha_{k,c}|)\right) & c \leq \min(z_k,\tilde{z}_k) \\ \frac{1}{2}\left(f(\tilde{z})-f(z)\right)\nabla_{\alpha_{k,c}}\log q_\theta(\tilde{b}_{k,c}) & z_k < c \leq \tilde{z}_k \\ \frac{1}{2}\left(f(z)-f(\tilde{z})\right)\nabla_{\alpha_{k,c}}\log q_\theta(b_{k,c}) & \tilde{z}_k < c \leq z_k \\ 0 & c > \max(z_k,\tilde{z}_k) \end{cases}.$$

We claim that for any $b_{-k,c}$ and $\tilde{b}_{-k,c}$,

$$\mathbb{E}_{b_{k,c},\tilde{b}_{k,c}}\left[g_{\text{DisARM-SB}\,k,c}\right] = \frac{1}{2}\mathbb{E}_{b_{k,c}}\left[f(h(b))\nabla_{\alpha_{k,c}}\log q_\theta(b_{k,c})\right] + \frac{1}{2}\mathbb{E}_{\tilde{b}_{k,c}}\left[f(h(\tilde{b}))\nabla_{\alpha_{k,c}}\log q_\theta(\tilde{b}_{k,c})\right],$$

which immediately implies unbiasedness. Importantly, the conditions defining the estimator can be determined solely based on $b_{-k,c}$ and $\tilde{b}_{-k,c}$, so it suffices to verify that the estimator is unbiased for each case separately. In the first case, $g_{\text{DisARM-SB}}$ is the DisARM estimator from (Dong et al., 2020) which was previously shown to be unbiased. The second and third cases are reminiscent of the 2-sample RLOO estimator, however, in the coupled case, such an estimator must be justified as unbiased. In the second case,

$$\mathbb{E}_{b_{k,c},\tilde{b}_{k,c}}\left[\frac{1}{2}\left(f(\tilde{z})-f(z)\right)\nabla_{\alpha_{k,c}}\log q_\theta(\tilde{b}_{k,c})\right]$$

$$= \frac{1}{2}\mathbb{E}_{\tilde{b}_{k,c}}\left[f(\tilde{z})\nabla_{\alpha_{k,c}}\log q_\theta(\tilde{b}_{k,c})\right] - \frac{1}{2}\mathbb{E}_{\tilde{b}_{k,c}}\left[\mathbb{E}_{b_{k,c}|\tilde{b}_{k,c}}[f(z)]\nabla_{\alpha_{k,c}}\log q_\theta(\tilde{b}_{k,c})\right]$$

$$= \frac{1}{2}\mathbb{E}_{\tilde{b}_{k,c}}\left[f(\tilde{z})\nabla_{\alpha_{k,c}}\log q_\theta(\tilde{b}_{k,c})\right] - \frac{1}{2}\overbrace{\mathbb{E}_{\tilde{b}_{k,c}}\left[f(z)\nabla_{\alpha_{k,c}}\log q_\theta(\tilde{b}_{k,c})\right]}^{0}$$

$$= \frac{1}{2}\mathbb{E}_{\tilde{b}_{k,c}}\left[f(\tilde{z})\nabla_{\alpha_{k,c}}\log q_\theta(\tilde{b}_{k,c})\right] + \frac{1}{2}\overbrace{\mathbb{E}_{b_{k,c}}\left[f(z)\nabla_{\alpha_{k,c}}\log q_\theta(b_{k,c})\right]}^{0}.$$

$f(z)$ depends on $\tilde{b}_{k,c}$ through the coupled sample $b_{k,c}$, however the condition $c > z_k$ implies that $z$ does not depend on $b_{k,c}$, so $f(z)$ is a constant with respect to both $b_{k,c}$ and $\tilde{b}_{k,c}$, resulting in the vanishing terms. The third case follows by symmetry. In the fourth case, the condition $c > \max(z_k,\tilde{z}_k)$ implies that neither $z$ nor $\tilde{z}$ depend on $b_{k,c}$ or $\tilde{b}_{k,c}$ hence the gradient vanishes.

### A.5 Tree-Structured Coupling

We construct a categorical sample based on a binary sequence arranged as a balanced binary tree. Considering a binary sequence $b = [b_0, b_1, b_2, \cdots, b_{C-1}]$, we interpret as a binary tree recursively with root $b[0]$ and left subtree $b[1 : len(b)//2 + 1]$ and right subtree $b[len(b)//2 + 1 :]$. The binary variables correspond to internal routing decisions in the binary tree with the categories as the leaves, so that $T(b)$ is defined by the following recursive function (assuming $C$ is a power of 2)

```
def T(b):
    half = len(b)//2 + 1
    if not b:
        return 1
    elif b[0] == 0:
```

```
        return  T(b[1:half])
    else:
        return  half + T(b[half:])
```

For each binary variable $b_i$, we can find all the categories residing in its left subtree (Left(i)) and all the categories in its right subtree Right(i). We would like the probability of traversing the right subtree $\sigma(\alpha_i)$ to be such that

$$\sigma(\alpha_i) = \frac{\sum_{j \in \text{Right(i)}} q(z = j)}{\sum_{j \in \text{Right(i)}} q(z = j) + \sum_{j \in \text{Left(i)}} q(z = j)}$$

Hence,

$$\alpha_i = \log \frac{\sum_{j \in \text{Right(i)}} q(z = j)}{\sum_{j \in \text{Left(i)}} q(z = j)}.$$

Recall, the estimator from the maintext. Given binary variables $b_1, \ldots, b_{C-1}$, let $I(b_1, \ldots, b_{C-1})$ be the set of variables used in routing decisions ($|I(b_1, \ldots, b_{C-1})| = \log_2 C$). With a pair of antithetically sampled binary sequences $(b, \tilde{b})$, the following is an unbiased estimator:

$$g_{\text{DisARM-Tree}\,k,c} = \begin{cases} \frac{1}{2}\left(f(z) - f(\tilde{z})\right)\left((-1)^{\tilde{b}_{k,c}} \mathbb{1}_{b_{k,c} \neq \tilde{b}_{k,c}} \sigma(|(\alpha_{k,c}|)\right) & c \in I(b_{k,.}) \cap I(\tilde{b}_{k,.}) \\ \frac{1}{2}\left(f(\tilde{z}) - f(z)\right)\nabla_{\alpha_{k,c}} \log q_\theta(\tilde{b}_{k,c}) & c \in I(\tilde{b}_{k,.}) - I(b_{k,.}) \\ \frac{1}{2}\left(f(z) - f(\tilde{z})\right)\nabla_{\alpha_{k,c}} \log q_\theta(b_{k,c}) & c \in I(b_{k,.}) - I(\tilde{b}_{k,.}) \\ 0 & c \notin I(b_{k,.}) \cup I(\tilde{b}_{k,.}) \end{cases}. \quad (6)$$

Closely following the argument from the previous section, we claim that for any $b_{-k,c}$ and $\tilde{b}_{-k,c}$,

$$\mathbb{E}_{b_{k,c}, \tilde{b}_{k,c}}\left[g_{\text{DisARM-Tree}\,k,c}\right] = \frac{1}{2}\mathbb{E}_{b_{k,c}}\left[f(h(b))\nabla_{\alpha_{k,c}} \log q_\theta(b_{k,c})\right] + \frac{1}{2}\mathbb{E}_{\tilde{b}_{k,c}}\left[f(h(\tilde{b}))\nabla_{\alpha_{k,c}} \log q_\theta(\tilde{b}_{k,c})\right],$$

which immediately implies unbiasedness. Importantly, the conditions defining the estimator can be determined solely based on $b_{-k,c}$ and $\tilde{b}_{-k,c}$, so it suffices to verify that the estimator is unbiased for each case separately. In the first case, $g_{\text{DisARM-Tree}}$ is the DisARM estimator from (Dong et al., 2020) which was previously shown to be unbiased. The second and third cases are reminiscent of the 2-sample RLOO estimator, however, in the coupled case, such an estimator must be justified as unbiased. In the second case, $f(z)$ depends on $\tilde{b}_{k,c}$ through the coupled sample $b_{k,c}$, however the condition $c \in I(\tilde{b}_{k,.}) - I(b_{k,.})$ implies that $z$ does not depend on $b_{k,c}$, so $f(z)$ is a constant with respect to both $b_{k,c}$ and $\tilde{b}_{k,c}$, resulting in the vanishing terms. Thus, we can apply the same argument as in the previous section. The third case follows by symmetry. In the fourth case, the condition $c \notin I(b_{k,.}) \cup I(\tilde{b}_{k,.})$ implies that neither $z$ nor $\tilde{z}$ depend on $b_{k,c}$ or $\tilde{b}_{k,c}$ hence the gradient vanishes.

### A.6 Rao-Blackwellized ARS & ARSM

First, we briefly review the Augment-REINFORCE-Swap (ARS) & Augment-REINFORCE-Swap-Merge (ARSM) estimators (Yin et al., 2019). Yin et al. (2019) use the fact that the discrete distribution can be reparameterized by an underlying continuous augmentation: if $\pi \sim \prod_k \text{Dirichlet}(1_C)$ and $z_k := \arg\min_i \pi_{k,i} e^{-\alpha_{k,i}}$, then $z_k \sim \text{Cat}(\alpha_k)$; and show that $\nabla_{\alpha_{k,c}} \mathbb{E}_{q_\theta(z)}\left[f(z)\right] = \mathbb{E}_\pi\left[f(z)(1 - C\pi_{k,c})\right]$. Furthermore, they define a swapped probability matrix $\pi_k^{i \leftrightarrows j}$ by swapping the entries at indices $i$ and $j$ in $\pi_k$

$$\pi_{k,c}^{i \leftrightarrows j} := \begin{cases} \pi_{k,i} & c = j \\ \pi_{k,j} & c = i \\ \pi_{k,c} & \text{o.w.} \end{cases},$$

and $z_k^{i \leftrightarrows j} := \arg\min_c \pi_{k,c}^{i \leftrightarrows j} e^{-\alpha_{k,c}}$. Using these constructions, they show an important identity

$$\nabla_{\alpha_{k,c}} \mathbb{E}_{q_\theta(z)}\left[f(z)\right] = \mathbb{E}_\pi\left[g_{\text{ARS}\,k,c} := \left[f(z^{c \leftrightarrows j}) - \frac{1}{C}\sum_{m=1}^{C} f(z^{m \leftrightarrows j})\right](1 - C\pi_{k,j})\right],$$

which shows that $g_{\mathrm{ARS}k,c}$ is an unbiased estimator. To further improve the estimator, Yin et al. (2019) average over the choice of the reference $j$, resulting in the ARSM estimator

$$g_{\mathrm{ARSM}k,c} := \frac{1}{C}\sum_{j=1}^{C}\left[f(z^{c \leftleftarrows j}) - \frac{1}{C}\sum_{m=1}^{C}f(z^{m \leftleftarrows j})\right](1 - C\pi_{k,j}).$$

Notably, both ARS and ARSM only evaluate $f$ at discrete values, and thus do not rely on a continuous relaxation.

### A.6.1 Rao-Blackwellization

Motivated by the approach of Dong et al. (2020), we can derive improved versions of ARS and ARSM by integrating out the extra randomness due to the continuous variables. ARS and ARSM heavily rely on a continuous reparameterization of the problem, yet the original problem only depends on the discrete values. Ideally, we would integrate out $\pi | z^{1 \leftleftarrows j}, ..., z^{C \leftleftarrows j}$, however, unlike in the binary case, computing the expectation analytically appears infeasible. Instead, we analytically integrate out two dimensions of $\pi$ and use Monte Carlo sampling to deal with the rest. This is a straightforward albeit tedious calculation

Starting with ARS, ideally, we would like to compute

$$\mathbb{E}_{\pi|z^{1 \leftleftarrows j},...,z^{C \leftleftarrows j}}\left[g_{\mathrm{ARS}k,c}\right] = \left[f(z^{c \leftleftarrows j}) - \frac{1}{C}\sum_{m=1}^{C}f(z^{m \leftleftarrows j})\right](1 - C\mathbb{E}_{\pi|z^{1 \leftleftarrows j},...,z^{C \leftleftarrows j}}\left[\pi_{k,j}\right])$$

$$= \left[f(z^{c \leftleftarrows j}) - \frac{1}{C}\sum_{m=1}^{C}f(z^{m \leftleftarrows j})\right](1 - C\mathbb{E}_{\pi_{k,j}|z_k^{1 \leftleftarrows j},...,z_k^{C \leftleftarrows j}}\left[\pi_{k,j}\right]),$$

taking advantage of independence between dimensions (indexed by $k$). To reduce notational clutter, we omit the dimension index in the following derivation.

We have reduced the problem to computing

$$\mathbb{E}_{\pi_j|z^{1 \leftleftarrows j},...,z^{C \leftleftarrows j}}\left[\pi_j\right],$$

but were unable to compute the expectation analytically. Instead, we analytically integrate out some dimensions of $\pi$ and use Monte Carlo sampling to deal with the rest. First, we know that $\sum_i \pi_i = 1$, so one variable is redundant (denote this choice by $l$). Next, we show how to integrate the reference index $j \neq l$ (i.e., compute $\mathbb{E}_{\pi_j|\pi_{-j,l},z^{1 \leftleftarrows j},...,z^{C \leftleftarrows j}}\left[\pi_j\right]$, where $\pi_{-j,l}$ denotes $\pi$ excluding its $j$-th and $l$-th elements.).

The known values of $\pi_{-j,l}, z^{1 \leftleftarrows j}, ..., z^{C \leftleftarrows j}$ imply lower and upper bounds on $\pi_j$. First, because $1 - \sum_{i \neq l}\pi_i = \pi_l \geq 0$, we conclude that $\pi_j \leq 1 - \sum_{i \neq j,l}\pi_i$. To determine the implications of the configurations $z^{1 \leftleftarrows j}, ..., z^{C \leftleftarrows j}$, it is helpful to define some additional notation. Let $s^{c \leftleftarrows j} := \pi^{c \leftleftarrows j}e^{-\alpha}$. Let's look at what the value of $z^{m \leftleftarrows j} := \arg\min_i s_i^{m \leftleftarrows j}$ tells us about $\pi_j$. We need to consider two cases:

- $z^{m \leftleftarrows j} = m$: This means that $s_m^{m \leftleftarrows j} = \pi_j e^{-\alpha_m}$ is the smallest entry in $s^{m \leftleftarrows j}$: $\pi_j e^{-\alpha_m} \leq \min_{i \neq m} s_i^{m \leftleftarrows j}$, which implies that $\pi_j \leq \min_{i \neq m} e^{\alpha_m} s_i^{m \leftleftarrows j}$.

  $e^{\alpha_m} s_i^{m \leftleftarrows j}$ contains $\pi_l$ when $m = l$ and $i = j$ or $m \neq l$ and $i = l$. When $m = l$ and $i = j$, we have that $e^{\alpha_l} s_j^{m \leftleftarrows j} = e^{\alpha_l}\pi_l e^{-\alpha_j} = (1 - \sum_{n \neq j,l}\pi_n - \pi_j)e^{\alpha_l - \alpha_j}$. Therefore,

  $$\pi_j \leq \frac{(1 - \sum_{n \neq j,l}\pi_n)e^{-\alpha_j}}{e^{-\alpha_j} + e^{-\alpha_l}}.$$

  A similar computation is required for the case $m \neq l$ and $i = l$.

- $z^{m \leftleftarrows j} \neq m$: This means that $\pi_j e^{-\alpha_m}$ is larger than the smallest entry in $s^{m \leftleftarrows j}$: $\pi_j e^{-\alpha_m} \geq \min_i s_i^{m \leftleftarrows j}$ which implies that $\pi_j \geq \min_i e^{\alpha_m} s_i^{m \leftleftarrows j}$. As above, we can eliminate $\pi_l$ from the bounds.

Finally, we aggregate the inequalities to compute the lower and upper bounds. Because $\pi \sim$ Dirichlet$(1_C)$ is a uniform distribution over the simplex, $\pi_j|\pi_{-j,l}, z^{1 \leftleftarrows j}, ..., z^{C \leftleftarrows j}$ will be uniformly

distributed over an interval, which means that it suffices to compute the lower and upper bounds to compute the expectation.

We can apply the same ideas to ARSM, however, in preliminary experiments with ARS+, we found that leveraging the symmetry (described next) was responsible for most of the performance improvement for ARS+. So, for ARSM, we reduce the variance only by leveraging the symmetry and call the resulting estimator ARSM+.

Furthermore, When all of the swapped $z$s agree on a dimension (i.e., $z_k^{1 \leftrightharpoons j} = \cdots = z_k^{C \leftrightharpoons j}$), then we will show that both $g_{\mathrm{ARS}k,c}$ and $g_{\mathrm{ARSM}k,c}$ vanish in expectation, so we can zero out these terms explicitly. The high level intuition is that even though they may disagree in other dimensions for a single sample because the other dimensions are independent and expectations are linear, in expectation they cancel out. Let $\delta_k = \mathbb{1}_{z_k^{1 \leftrightharpoons j} = \cdots = z_k^{C \leftrightharpoons j}}$. Then, we have

$$
\mathbb{E}_{\pi|\delta_k=1}\left[g_{\mathrm{ARS}k,c}\right] = \mathbb{E}_{\pi|\delta_k}\left[\left[f(z^{c \leftrightharpoons j}) - \frac{1}{C}\sum_{m=1}^{C} f(z^{m \leftrightharpoons j})\right](1 - C\pi_{k,j})\right]
$$

$$
= \mathbb{E}_{\pi_k|\delta_k=1}\left[\left(\mathbb{E}_{\pi_{-k}}\left[f(z^{c \leftrightharpoons j})\right] - \frac{1}{C}\sum_{m=1}^{C}\mathbb{E}_{\pi_{-k}}\left[f(z^{m \leftrightharpoons j})\right]\right)(1 - C\pi_{k,j})\right].
$$

Now, we claim that inside the expectation $\mathbb{E}_{\pi_{-k}}\left[f(z^{m \leftrightharpoons j})\right]$ is constant with respect to $m$. First, we know that inside the expectation $z_k^{1 \leftrightharpoons j} = \cdots = z_k^{C \leftrightharpoons j}$ and that the dimensions indexed by $k$ are independent. Because $\pi \sim \prod_k \mathrm{Dirichlet}(1_C)$, $\mathrm{Dirichlet}(1_C)$ is symmetric, and we are taking an unconditional expectation over the remaining dimensions, the value is invariant to the swapping operation. As a result, the entire expression vanishes. Thus, we conclude that

$$
(1 - \delta_k)g_{\mathrm{ARS}k,c}
$$

is still an unbiased estimator. A similar argument holds for $g_{\mathrm{ARSM}}$. This is complementary to the approach in the previous subsection and can done in combination

$$
g_{\mathrm{ARS}+k,c} := \mathbb{E}_{\pi_{k,j}|\pi_{k,-jl}, z_k^{1 \leftrightharpoons j}, \ldots, z_k^{C \leftrightharpoons j}}\left[(1 - \delta_k)g_{\mathrm{ARS}k,c}\right],
$$

where we choose $l \neq j$ uniformly at random. This is the estimator we use in our experiments.

### A.6.2 Evaluating Rao-Blackwellized ARS & ARSM

We train models with 10/5/3/2-category latent variables on dynamically binarized MNIST. For comparison, we train models with ARS, ARSM, and an $n$-sample RLOO. To match computation, RLOO uses $C$ samples for comparing against ARS/ARS+, and uses $C(C-1)/2+1$ samples for ARSM/ARSM+, where $C$ is the number of categories. Based on preliminary experiments with ARS+, we found that leveraging the symmetry led to most of the performance improvement for ARS+. So, for ARSM, we reduce the variance only by leveraging the symmetry and call the resulting estimator ARSM+.

As shown in Figure 8 and Appendix Figure 9, the proposed estimators, ARS+/ARSM+, significantly outperform ARS/ARSM. Surprisingly, we find that both ARS and ARSM underperform the simpler RLOO baseline in all cases. For $C = 2$, ARS+ and ARSM+ reduce to DisARM/U2G and as expected, outperform REINFORCE LOO; however, for $C > 2$, REINFORCE LOO is superior and the gap increases as $C$ does. This suggests that partially integrating out the randomness is insufficient to account for the variance introduced by the continuous augmentation.

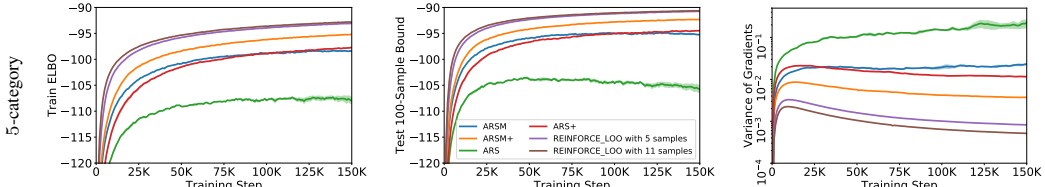

Figure 8: Training a non-linear categorical VAE with latent variables with 5 categories on dynamically binarized MNIST dataset by maximizing the ELBO. We plot the train ELBO (left column), test 100-sample bound (middle column), and the variance of gradient estimator (right column). We plot the mean and one standard error based on 5 runs from different random initializations.

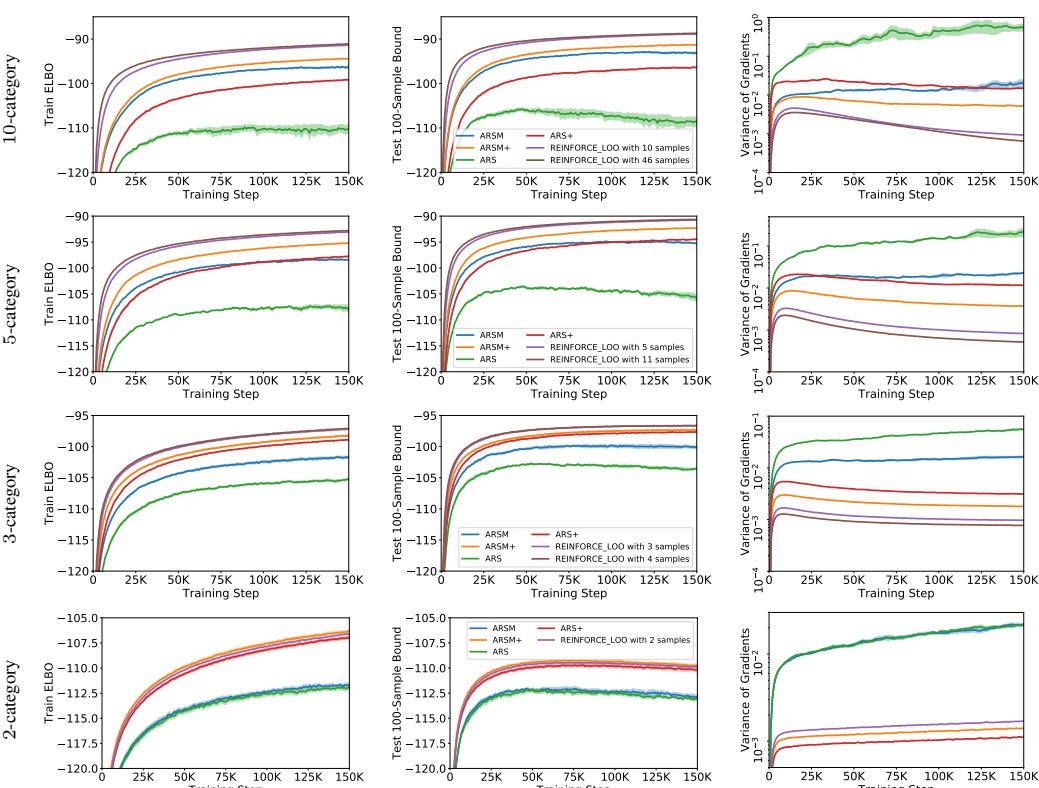

Figure 9: Training a non-linear categorical VAE with latent variables with 10/5/3/2 categories on dynamically binarized MNIST dataset by maximizing the ELBO. We plot the train ELBO (left column), the test 100-sample bound (middle column), and the variance of gradient estimator (right column). We plot the mean and one standard error based on 5 runs from different random initializations.