# OpenReview forum: "Coupled Gradient Estimators for Discrete Latent Variables"
_NeurIPS.cc/2021/Conference — NeurIPS 2021 Poster_

### Official Review · Reviewer_AbN2 · 2021-07-16

**Rating:** 6
**Confidence:** 3

**Summary:**

This paper considers the problem of designing a gradient estimator for categorical latent variables that is unbiased and does not depend on continuous relaxation. It proposes three different types of such estimators (called DisARM-IW, DisARM-{SB,Tree}, and {ARS,ARSM}+). DisARM-IW is based on a new derivation, which uses a coupling, of a previous estimator called DisARM; DisARM-{SB,Tree} are obtained by reparameterizing with binary variables and then applying DisARM; and {ARS,ARSM}+ are obtained by Rao-Blackwellizing previous estimators called {ARS,ARSM}. DisARM-{IW,SB,Tree} are shown to outperform a state-of-the-art estimator called RLOO on several benchmarks.


**Limitations And Societal Impact:**

Limitations and societal impact are discussed in the paper.

**Main Review:**

## Summary of Review

The paper considers an important problem and proposes new methods, some of which outperform the state-of-the-arts. One of the proposed methods (DisARM-IW) looks novel to me, but the other methods look less novel. The paper is technically sound. The paper needs more improvement in writing.

## Strengths

(1) This paper considers an important problem and proposes several solutions. Among the proposed solutions, DisARM-IW and DisARM-{SB,Tree} are shown to outperform a state-of-the-art estimator.

(2) DisARM-IW is novel to me in that its derivation looks new and it can be seen as a generalization of a previous estimator called DisARM.

## Weaknesses

(1) The estimators other than DisARM-IW look incremental and much less novel to me. DisARM-{SB,Tree} are obtained by applying reparameterization based on the stick-breaking or tree construction and then simply passing the reparameterized form to the well-known estimator DisARM. {ARS,ARSM}+ are obtained by simply following an approach similar to DisARM. I think little novelty in DisARM-{SB,Tree} is less problematic as they outperform RLOO.

(2) The writing needs more improvement. First, even though an antithetic Bernoulli coupling is used importantly in several parts of this paper, it is not explained at all in detail. Second, it was difficult to grasp the overall picture of, and connections between, proposed estimators. I was misled that DisARM-IW and DisARM-{SB,Tree} are similar types of estimators, which is not the case: the former does not use DisARM at all, while the latter uses DisARM importantly. In addition I felt that {ARS,ARSM}+ and Sec 3.2 are completely separate from the other estimators DisARM-* and Sec 3-3.1. This makes the paper less coherent and makes it difficult to understand the main point of the paper.

## Questions and comments

- Line 127-128. p(b_k,i) > 0.5 should be p(b_k,i=1) >= 0.5 (note >= instead of >)? If this is the case, isn’t it a problem if q(z_k=i) / \sum_{j=i}^C q(z_k=j) = 0.5 (Line 131), as this implies p(z_k>i, \tilde{z_k}>i) = 0 and violates the support condition?
- Line 106. Include the proof or calculation (to text or appendix) showing that Eq (3) recovers DisARM for a particular coupling, as this is one of the major contributions of this paper.

- Line 78-79. \sigma is undefined.
- Line 86-87. j is undefined.
- Line 133. \sum_{j=i+1}^C … should be \sum{j=i}^C?

-----
**Updates after the author response.** Thank the authors for answering my questions. The response gives a more clarified view of the paper. I am still positive on the paper and will keep the same score.

**Time Spent Reviewing:**

8

---

> ### Author Response · Authors · 2021-08-11
> **RE: Official Review**
>
> Thank you for your detailed comments.
>
> >Novelty ​​DisARM-{SB,Tree}
>
> Part of the novelty of these estimators is that they improve over the direct application of DisARM to a binary reparameterization by including an additional Rao-Blackwellization step after reparameterization that takes advantage of the structure of the problem. This corresponds to the last 3 lines in Eq. 4 and 5.
>
> >Relationship between DisARM-IW, DisARM-{SB,Tree}, and ARS+/ARSM+
>
> We will clarify the relationship between the estimators. All of these estimators are generalizations of DisARM, even though they are derived in very different ways. DisARM-IW reduces exactly to the original DisARM/U2G when using Bernoulli variables and an antithetic coupling. Although we mentioned this in the maintext, we will provide a derivation of this fact in the appendix.
>
> We agree that the structure of the paper with ARS+/ARSM+ was not great. We chose to include this estimator because it is the most direct generalization of DisARM to the Categorical setting (and our first attempt) and we thought the fact that it underperformed RLOO was of interest. In retrospect, it would be better to summarize the result and move the details to the appendix. We will do this in the next revision.
>
> >Defining antithetic Bernoulli coupling
>
> We will define the antithetic Bernoulli coupling explicitly before using it. Thank you for spotting this.
>
> > Line 127-128. p(b_k,i) > 0.5 should be p(b_k,i=1) >= 0.5 (note >= instead of >)? If this is the case, isn’t it a problem if q(z_k=i) / \sum_{j=i}^C q(z_k=j) = 0.5 (Line 131), as this implies p(z_k>i, \tilde{z_k}>i) = 0 and violates the support condition?
>
> Yes, we will amend this.
>
> > Line 133. \sum_{j=i+1}^C … should be \sum{j=i}^C?
>
> Yes, thank you for pointing out this typo in the text. The experiments implement this correctly.

---

### Official Review · Reviewer_u8r7 · 2021-07-16

**Rating:** 4
**Confidence:** 4

**Summary:**

The paper proposes three versions of DisARM/U2G estimators.
The authors extend the binary version to the categorical random variables, and the extentions are in the importance sampling sense (IW) and stick-breaking coupling perspecitve (SB and Tree).
They also verify their method through VAE experiments.

**Limitations And Societal Impact:**

The authors argue that they pointed out the limitation of their work in the conclusion (but I can not find it), and there is no negative societal impact.

**Main Review:**

Originality:
Extending the idea of binary random variable gradient estimator to the categorical case is natural.
Also, constructing categorical variable through stick-breaking or tree construction is also a natural idea.
However, combining those two ideas in the research on the gradient estimotors for the categorical variable seems novel.

Quality:
The paper seems technically sound.
The appendix substantiate the unbiasness of the proposed gradient estimators.

Clarity:
I can't say that the paper is well-written (it seems that it can be written in some better way), the paper is written in relatively clear manner.

Significance:
My concern on this paper is in two parts.
The first one is that the experiment is too limited.
The baselines are limited and the experimental settings, where the reparameterization estimators are excluded and a lot of score function estimators are missing. (including UnOrd [1])
Also, the experimental task is limited to VAE case.
As the research on the gradients over categorical random variables grows, more extreme experiments are gradually required to differentiate the effect of the proposed methods.
I suggest authors demonstrate the idea in the broader experimental settings.
The second concern is that the method (or the implementation) is more complex than the sole baseline, RLOO, but the performance gap is too marginal.
I guess the authors should provide an example that the proposed complex method gives a clear benefit compared to the baseline.
Therefore, I lean my score to reject.

Questions:
- How does the running time of each model as the number of latent categories changes? (compared to RLOO) Especially, the stick-breaking construction might take a long time.
- From the learning curves in Fig 1 and 2, it seems that the models are not reached the optimal point, and it seems they still have a learning chance. Am I missing something?
- From the right-most figures of Fig 1, the variance of gradients reduces up to a certain point and keeps increases later. What causes such curve shapes?
- How does the performance change (including RLOO) if one sets the extreme prior in VAE?

[1] Estimating gradients for discrete random variables by sampling without replacement., Kool et al., ICLR 2020.


**Time Spent Reviewing:**

8

---

> ### Author Response · Authors · 2021-08-11
> **RE: Official Review**
>
> Thank you for your review.
>
> > [...] the experiment is too limited [...]
>
> We used RLOO because it is a strong baseline that has been shown to be as least as good as the more complex methods such as RELAX/REBAR/UnOrd in the high-dimensional setting we are interested in (see [1] and [2]). We do not compare to the reparameterization-trick because it is not directly applicable to discrete latent variables and methods like Concrete/Gumbel-softmax relaxations produce biased gradients. Training a VAE with discrete latent variables on the three datasets we use is a standard setting used by previous work and the gains are comparable to previous works (e.g., [3-10]).
>
> > [...] the method (or the implementation) is more complex than the sole baseline, RLOO, but the performance gap is too marginal [...]
>
> While our methods are indeed somewhat more complex than RLOO, we provide an open source implementation of them to make them easier to apply.
>
> > How does the running time of each model as the number of latent categories changes? (compared to RLOO) Especially, the stick-breaking construction might take a long time.
>
> All our methods have the same computational complexity as RLOO. The stick-breaking construction computation is dwarfed by the (relatively) expensive neural network evaluations.
>
> > From the learning curves in Fig 1 and 2, it seems that the models are not reached the optimal point, and it seems they still have a learning chance. Am I missing something?
>
> As long as the same settings are used for all methods, we found that the qualitative results are similar. For example, training for longer (1 million steps)
>
> ELBO     | DisARM-IW | DisARM-Tree |DisARM-SB | RLOO
> ---------|-----------|-------------|-----------------|-----
> Dynamic-MNIST  |-101.09&pm;0.12 |-100.67&pm;0.26 |-101.04&pm;0.14 |-101.84&pm;0.17
> FashionMNIST  |-238.45&pm;0.17 |-238.38&pm;0.26 |-237.98&pm;0.25 |-239.54&pm;0.31
> Omniglot |-118.34&pm;0.28 |-118.93&pm;0.26 |-118.23&pm;0.14 |-118.67&pm;0.79
>
> > From the right-most figures of Fig 1, the variance of gradients reduces up to a certain point and keeps increases later. What causes such curve shapes?
>
> The variance depends on the model parameters which change throughout training. At a high level, the model becomes more deterministic as training proceeds, however, fully understanding the dynamics of the training parameters and its impact on gradient variance is an interesting future direction.
>
> > How does the performance change (including RLOO) if one sets the extreme prior in VAE?
>
> Could you clarify what you mean by the "extreme prior"?
>
> Please see our global comment above covering the limitations of our work.
>
> References
> 1. Richter et al. "VarGrad: a low-variance gradient estimator for variational inference.", NeurIPS 2020
> 2. Kool at al. "Estimating Gradients for Discrete Random Variables by Sampling without Replacement", ICLR 2020
> 3. Mnih and Gregor., “Neural Variational Inference and Learning in Belief Networks.”, ICML 2014.
> 4. Mnih and Rezende., “Variational Inference for Monte Carlo Objectives.”, ICML 2016.
> 5. Maddison et al., “The Concrete Distribution: A Continuous Relaxation of Discrete Random Variables.”, ICLR 2016.
> 6. Jang et al., “Categorical reparameterization with gumbel-softmax.”, ICLR 2016.
> 7. Grathwohl et al. “Backpropagation through the void: Optimizing control variates for black-box gradient estimation.”, ICLR 2018.
> 8. Yin and Zhou., “ARM: Augment-REINFORCE-merge gradient for stochastic binary networks.”, ICLR 2019.
> 9. Dong et al., “DisARM: An Antithetic Gradient Estimator for Binary Latent Variables.”, NeurIPS 2020.
> 10. Dimitriev and Zhou., “ARMS: Antithetic-REINFORCE-Multi-Sample Gradient for Binary Variables.”, ICML 2021.

---

### Official Review · Reviewer_Msvo · 2021-07-16

**Rating:** 6
**Confidence:** 4

**Summary:**

The paper reformulates the DisARM/U2G (Dong et al., 2020; Yin et al., 2020) gradient estimators using importance sampling. While these estimators are originally applicable to binary latent variables, the new formulation allows for the extension to the categorical latent variables case by making use of couplings (common random numbers). The paper also shows that another family of estimators ARS & ARSM (Yin et al., 2019) is not performant compared to RLOO (leave-one-out REINFORCE) (Kool, et. al, 2009) even when its variance is reduced further with Rao-Blackwellization.


**Limitations And Societal Impact:**

There was not much discussion of the technical limitations of this work in the manuscript. It would be great if the authors can comment on those, especially if there is any extra computational overhead.

The potential societal impact of this work was sufficiently considered.

**Main Review:**

**Originality:** While the paper builds on previous work, the reformulation of DisARM/U2G (Dong et al., 2020; Yin et al., 2020) with importance sampling and the use of couplings are novel. The experimental evaluation is standard for this type of problem and uses the usual datasets and models. The paper does a decent job at positioning its contributions within the literature and differentiates itself from its direct comparisons; however, it still misses some relevant references, e.g. Lee et al. 2018, Peters and Welling 2018, Shayer et al. 2018, Cong et al. 2019, Richter, et. al. 2020.

**Quality:** I think this work is technically solid. The mathematical derivations seem to be correct to me (although I did not verify all of them in detail) and the motivation was clear. Although, I think that the paper is lacking some more explanation and intuition on why this type of estimator reduces the variance (I had to go to Owen 2009 to get some intuition).

One thing I was not able to verify is how computationally expensive this estimator is? In the experiments section, the authors state that: “These estimators require only 2 expensive function evaluations regardless of the number of categories (C) ”. How expensive is this compared to the 2-sample RLOO estimator?  Does the SB and Tree coupling constructions have any additional overhead? I think these questions should be addressed in the experimental section. Currently, the experiments only show the ELBO with respect to the training steps. I think a similar comparison needs to be performed but with respect to wall-clock time to show the effectiveness of this estimator in practice.

Furthermore, it is also good to include a comparison to other popular gradient estimators for categorical variables such as REBAR (Tucker, et. al. 2017) and RELAX (Grathwohl, et. al. 2018). While these estimators are expected to perform well because of their adaptivity, they are quite expensive to compute. I think that benchmarking w.r.t. wall-clock time might reveal that DisARM-IW is competitive in comparison.

Finally, I found the benchmarking of ARS+ & ARSM+ separately to be strange, since the main thesis of this paper is that these do not close the gap to RLOO motivating the need for DisARM-IW. I think the comparison with DisARM needs to be direct in this instance. Please see further comments below on this point.

**Clarity:** I think the paper is well-written and the technical details are mostly clear. However, I found the discussion of ARS & ARSM a little bit arbitrary. These estimators are only considered in the experiments and in section 3.2 on their Rao-Blackwellization.

Further, these are not compared to the contributed estimator (DisARM-IW/SB/Tree). I understand that their inclusion is to demonstrate they are not as effective in reducing the variance as the proposed estimator or RLOO, but couldn’t this comparison be made directly? At present, this paper reads as two distinct contributions (A major one: DisARM-IW and a minor one: Rao-Blackwellization of ARS & ARSM) in a single manuscript. I think the manuscript will be much clearer if the discussion of ARS & ARSM is integrated better in the exposition (including the experiments). Alternatively, this discussion could be demoted to the appendix or submitted as a workshop paper in the future.

Since this paper is relatively heavy on notation, I would suggest the following three points to improve its clarity:
* Needed: An algorithmic description (pseudocode) of DisARM-IW and the computation of the different couplings (some of this can be added to the supplement).
* Nice to have: following the standard notation of representing vectors in boldface.
* Nice to have: definitions of undefined symbols, e.g. $1_C$.

**Significance:** I think this paper provides a significant contribution to a few ML communities (mainly probabilistic modelling and RL). The introduction of the idea of coupling for variance reduction is an important contribution to these communities and can create many avenues for more research into variance reduction for Variational Inference and Policy Gradients. However, the impact of this paper is hampered by the limited experimental evaluation. That being said, I am willing to reconsider my evaluation based on the authors’ rebuttal/clarifications and the discussion with the other reviewers.


**Time Spent Reviewing:**

4

---

> ### Author Response · Authors · 2021-08-11
> **RE: Official Review**
>
> Thank you for your detailed comments.
>
> > the paper is lacking some more explanation and intuition on why this type of estimator reduces the variance (I had to go to Owen 2009 to get some intuition).
>
> Thank you for the suggestion. We will add intuition on variance reduction in the next revision.
>
> > One thing I was not able to verify is how computationally expensive this estimator is?
>
> The RLOO uses the same number of expensive function computations (e.g., NN evaluation). There is a small amount of numerical overhead with SB and Tree couplings, however, it is dwarfed by the (relatively) expensive NN evaluations, resulting in similar computation time for all methods.
>
> > Furthermore, it is also good to include a comparison to other popular gradient estimators for categorical variables such as REBAR (Tucker, et. al. 2017) and RELAX (Grathwohl, et. al. 2018). While these estimators are expected to perform well because of their adaptivity, they are quite expensive to compute. I think that benchmarking w.r.t. wall-clock time might reveal that DisARM-IW is competitive in comparison.
>
> Thank you for the suggestion. We used RLOO because it is a strong baseline that has been shown to be competitive with more complex methods such as RELAX/REBAR in the setting we evaluated (see [1]). In addition, they rely on continuous relaxations which preclude conditional computation applications. For completeness, we are running these experiments and will add those results shortly.
>
> > [Rao-Blackwellization of ARS & ARSM]
>
> Yes, we agree that the integration of these results did not work as well as we had hoped. In retrospect, it would be better to summarize the result and move the details to the appendix as you suggest. We will do this in the next revision.
>
> Thank you for the clarity suggestions; we will incorporate them.
>
> References
> 1. Richter et al. "VarGrad: a low-variance gradient estimator for variational inference.", NeurIPS 2020.

---

> > ### Comment · Reviewer_Msvo · 2021-09-01
> > **Reponse to authors**
> >
> > Thank you for your response. I have read the rebuttal and taken it into account in the discussion with the other reviewers.

---

### Official Review · Reviewer_ozdD · 2021-07-20

**Rating:** 5
**Confidence:** 3

**Summary:**

The authors consider optimization problems that can be formulated as maximizing parameterized functions that can be expressed as an expectation of a function f with respect to some factorial catergorical distribution.  These kinds of optimization problems arise naturally in standard mean-field ELBO optimization problems.  The proposed approaches aim to go beyond existing work by introducing a stick-breaking based coupling technique to reduce the variance of the sampling based estimator.  This is done by first presenting an importance-sampling characterization of DisARM, from which the newly proposed techniques are derived.

**Limitations And Societal Impact:**

In my opinion, no detailed discussion of limitations is provided.

**Main Review:**

Originality:  While the work does build on top of existing work, the perspective is new.

Quality:  1) The claim that, "In the case of nonlinear models (Figure 1), all three proposed estimators perform similarly, with lower
240 gradient variance and better performance than the baseline estimator (RLOO) across all datasets," does not seem to match the figures, e.g., not on Omniglot.

2) The data sets selected for comparison are all quite similar.  It is unclear to me that similar behavioral trends would hold for a wider variety of data sets.  Can the authors explain if they looked at other data sets and why these data sets were chosen?

3) The Rao-blackwellized portion of the experiment section feels like a bit of an afterthought and most of it is relegated to the appendix.  I think I'd prefer a more detailed study of disarm variants.

Clarity:  1) While this paper is a bit out of area for me, it is incredibly difficult to read as many assertions are made without concreate explanations or citations of why something is true, e.g., "Monte Carlo estimates of the first term can have large variance."  This should have a citation.  There are many more examples of this, too many to reasonably include here which greatly limited my ability to evaluate and certify the results.

2) The whole discussion of related work seems like a bit of a mess to me as the paper bounces back and forth from discussing related work, a background section on related work, a section that builds on related work, and then a separate related work section. I think these sections can be combined and streamlined so as to minimize repetition of information and allow for additional explanations where needed.

Significance:  The performance improvements of the new approach range from mild to moderate, but the perspective may be of interest more broadly.

**Time Spent Reviewing:**

3 hours

---

> ### Author Response · Authors · 2021-08-11
> **RE: Official Review**
>
> Thank you for your review.
>
> >The choice of datasets.
>
> We used the datasets commonly chosen as benchmarks for papers on unbiased gradient estimation for discrete latent variables. The same datasets were used in for example [1-8].
>
> >Clarity.
>
> Thank you for raising this issue; the clarity can certainly be improved, and we will try to address these points in the next revision.
>
> >Rao-Blackwellized estimator
>
> Thank you for the suggestion. We chose to include this estimator because it is the most direct generalization of DisARM to the Categorical setting and we thought the fact that it underperformed RLOO was of interest. In retrospect, it would be better to summarize the result and move the details to the appendix as you suggest. We will do this in the next revision.
>
> > The claim that, "In the case of nonlinear models (Figure 1), all three proposed estimators perform similarly, with lower 240 gradient variance and better performance than the baseline estimator (RLOO) across all datasets," does not seem to match the figures, e.g., not on Omniglot.
>
> Thank you for pointing out this inconsistency. We have revised the text to match the actual performance.
>
> > Related work clarity
>
> The related work section can indeed be better integrated with the rest of the paper, so we will streamline it in the next revision.
>
> Please see our global comment above covering the limitations of our work.
>
> References
> 1. Mnih and Gregor., “Neural Variational Inference and Learning in Belief Networks.”, ICML 2014.
> 2. Mnih and Rezende., “Variational Inference for Monte Carlo Objectives.”, ICML 2016.
> 3. Maddison et al., “The Concrete Distribution: A Continuous Relaxation of Discrete Random Variables.”, ICLR 2016.
> 4. Jang et al., “Categorical reparameterization with gumbel-softmax.”, ICLR 2016.
> 5. Grathwohl et al. “Backpropagation through the void: Optimizing control variates for black-box gradient estimation.”, ICLR 2018.
> 6. Yin and Zhou., “ARM: Augment-REINFORCE-merge gradient for stochastic binary networks.”, ICLR 2019.
> 7. Dong et al., “DisARM: An Antithetic Gradient Estimator for Binary Latent Variables.”, NeurIPS 2020.
> 8. Dimitriev and Zhou., “ARMS: Antithetic-REINFORCE-Multi-Sample Gradient for Binary Variables.”, ICML 2021.

---

> > ### Comment · Reviewer_ozdD · 2021-09-01
> > **Thanks**
> >
> > Thanks for your clarifications.

---

### Official Review · Reviewer_KiRx · 2021-08-25

**Rating:** 6
**Confidence:** 4

**Summary:**

This problem addresses the problem of differentiating expectations of discrete random variables. The authors proceed from a derivation of DisARM that applies importance sampling to the RLOO gradient, extending this estimator beyond 2 categories. A core component of this new multi-sample gradient estimator is a coupling which leads to (negatively) correlated samples of the categorical. The main proposed coupling uses stick-breaking. The authors propose two estimators that do not use importance sampling, but extend DisARM to the categorical case by reparametrising a categorical using binary random variables. This approach is amenable to standard DisARM. The authors consider two such binarizations: first using the stick-breaking construction again, second using a decision tree. Finally, the authors propose ARS+ and ARSM+ plus, which are Rao-Blackwellized versions of ARS and ARSM. Experimental evaluation focuses on categorical latent variable VAEs.

**Limitations And Societal Impact:**

I think the authors have addressed wider societal impact adequately. On limitations, see my main review.

**Main Review:**

Novelty/related work
-----------------------------
The work constitutes a natural extension of existing work, focusing on moving from binary to categorical random variables. Regarding related work, a piece that seems to be missing is ARMS [1]: a recent (ICML 2021) paper that generalizes RLOO and DisARM, but remains focused on the binary case. To me, the key mathematical contributions seem to be
1. Extending antithetic coupling based methods to categoricals (DisARM-IW)
2. Proposing an explicit coupling using stick breaking (DisARM-IW)
3. Alternatively, two binarizations of categoricals that can then use vanilla DisARM (-SB and -Tree)
4. Rao-Blackwellization of ARS and ARSM to give ARS+ and ARSM+

With all these different variations floating around, I do think the authors should do a little bit more work to clarify the relations between these different methods. For instance, it wasn’t 100% clear to me whether there is any relationship between ARS+ and ARSM+ and the DisARM-* methods. My intuition was no, because the DisARM-* methods don’t use any continuous augmentation that could be integrated out. I think clarifying this would be very helpful, perhaps with some kind of table.

Quality
---------
The paper is of good quality. Although I haven’t read into the appendix in great detail, I have good confidence in the underlying mathematics of the approach. One thing that could have further grounded this work is stronger theoretical analysis. When considering the properties of a good coupling, the authors say (112-114) “moving mass away from this configuration will reduce variance. Furthermore, for the estimator to be valid, the coupling must put non-zero mass on all $z_k \ne \tilde{z}_k$ configurations.” But this is not followed up by more careful analysis. For instance, the variance of the new estimator is not calculated / bounded. Furthermore in 279-280 “DisARM-IW is simpler to implement, more natural to understand, and easier to generalize, we recommend it in practice”. This kind of comparison between different proposed methods would again have been stronger with some kind of analysis or calculation of the actual variance, to guide a selection of the right estimator.

On the empirical side, I would say the authors have been fairly detailed here. They have looked at three sensible datasets, and compared against suitable baselines. It was a tiny bit confusing to separate out the comparison as DisARM-* versus RLOO, and secondarily RLOO versus ARS(+) and ARSM(+). But I think the logic makes sense here: the estimators in the first group only require 2 function evaluations. One direction that the experiments could have been pushed is to consider other models beyond the VAE. Could these estimators be applied to belief nets, or would be number of discrete latents be problematic?

On strengths and weaknesses, one thing that could be discussed in more detail is the relative computational cost of all of the estimators being discussed in this paper (as there are many). For the DisARM-Tree estimator, is there any cost to calculating the tree probabilities (if C was extremely large, perhaps this becomes problematic)?

Clarity
---------
I found the paper to be well-written. However, there are some places where I think the authors could be clearer. As mentioned previously, a more precise and self-contained comparison between the different estimators proposed in this paper (plus some baselines) would help. I understand that this information is present in the paper, but it can be hard to locate. For instance, a table with concrete information such as complexity, continuous augmentation (y/n) and so on could be a real asset. One thing that should be stressed is the distinction between DisARM-IW with a stick breaking coupling and DisARM-SB, there is definitely a possibility for confusion here.

Significance
----------------
This paper deals with an important area of ML research. It makes some sensible and logical next steps in the development of low variance estimators for discrete random variables. The empirical gains are not colossal, but they do provide new state of the art performance. I would say this paper is definitely a valuable contribution, without being a huge leap forward.

[1] Dimitriev, Alek, and Mingyuan Zhou. "ARMS: Antithetic-REINFORCE-Multi-Sample Gradient for Binary Variables." arXiv preprint arXiv:2105.14141 (2021).

**Time Spent Reviewing:**

4

---

> ### Author Response · Authors · 2021-08-27
> **RE: Official Review**
>
> Thank you for your detailed comments.
>
> > Regarding related work, a piece that seems to be missing is ARMS [1]: a recent (ICML 2021) paper that generalizes RLOO and DisARM, but remains focused on the binary case.
>
> Thank you for pointing this out. We will include this in our next revision with a discussion of its relation to our work. We do note that this paper appeared on arXiv after the NeurIPS submission deadline.
>
> Relatedly, we extended our work to the multisample case. As noted in ARMS, the multisample RLOO estimator is the average of all pairs of the 2-sample RLOO estimator. Given n independently sampled coupled pairs, the 2n RLOO estimator can be written as the average over *all* pairs. For most pairs, the two samples come from different coupled pairs (hence the samples are independent), so the 2-sample RLOO estimator will be unbiased. However, for n choices, the pair is a coupled pair. To correct for this bias, we can simply replace those terms with a DisARM-* estimator which is unbiased for coupled pairs.
>
> We ran experiments with 5 antithetic pairs (10 samples) and 10 antithetic pairs (20 samples) and found that the proposed estimators outperform RLOO with a comparable number of independent samples. Even with the increasing number of samples, the performance improvement of the proposed estimators w.r.t. RLOO still holds.
>
> * **10 Samples/5 Pairs**
>
> ELBO    |  DisARM-Tree    |  DisARM-SB      |   DisARM-IW     |        RLOO
> --------|-----------------|-----------------|-----------------|----------------
> dynamic | -91.36&pm;0.11  | -91.67&pm;0.08  | -91.85&pm;0.04  | -92.35&pm;0.10
> fashion | -231.01&pm;0.14 | -231.38&pm;0.14 | -231.14&pm;0.17 | -231.78&pm;0.16
> omniglot| -109.00&pm;0.10 | -108.89&pm;0.10 | -108.79&pm;0.14 | -109.73&pm;0.07
>
> * **20 Samples/10 Pairs**
>
> ELBO    |  DisARM-Tree    |  DisARM-SB      |   DisARM-IW     |        RLOO
> --------|-----------------|-----------------|-----------------|----------------
> dynamic | -90.80&pm;0.07  | -91.01&pm;0.06  | -90.95&pm;0.08  | -91.86&pm;0.15
> fashion | -230.95&pm;0.21 | -231.21&pm;0.18 | -231.25&pm;0.29 | -231.59&pm;0.21
> omniglot| -108.25&pm;0.03 | -108.15&pm;0.06 | -108.08&pm;0.06 | -108.95&pm;0.17
>
> > With all these different variations floating around, I do think the authors should do a little bit more work to clarify the relations between these different methods. For instance, it wasn’t 100% clear to me whether there is any relationship between ARS+ and ARSM+ and the DisARM-* methods. My intuition was no, because the DisARM-* methods don’t use any continuous augmentation that could be integrated out. I think clarifying this would be very helpful, perhaps with some kind of table.
>
> Thank you for the suggestion. We agree that such a table would be very helpful, and we will add a table clarifying the relationships between the estimators and their properties. You are correct that we do not know of a direct relationship between ARS+/ARSM+ and DisARM-* methods.
>
> > One thing that could have further grounded this work is stronger theoretical analysis. When considering the properties of a good coupling, the authors say (112-114) “moving mass away from this configuration will reduce variance. Furthermore, for the estimator to be valid, the coupling must put non-zero mass on all
>  configurations.” But this is not followed up by more careful analysis. For instance, the variance of the new estimator is not calculated / bounded. Furthermore in 279-280 “DisARM-IW is simpler to implement, more natural to understand, and easier to generalize, we recommend it in practice”. This kind of comparison between different proposed methods would again have been stronger with some kind of analysis or calculation of the actual variance, to guide a selection of the right estimator.
>
> Agreed, the challenge we ran into is that quantifying the variance analytically was in terms of statistical quantities of $f$, which we did not assume any structure on. However, you are right that it would still be helpful to the reader and further to detail some situations where we make assumptions on $f$ which guided our decisions. We will add a discussion of this in the next revision.
>
> We do note that we can construct a parameterized coupling that smoothly interpolates between the independent coupling and the antithetic Bernoulli (e.g., Appendix B of DisARM), which we can use to construct categorical couplings (as our couplings are based on the Bernoulli couplings) interpolating between the independent coupling and the “antithetic” coupling. DisARM-* can be extended to use these couplings in which case it reduces to RLOO with the independent coupling and is unbiased for any setting of the coupling parameter. As a result, if the coupling parameter is set optimally to minimize variance, we could guarantee no worse variance than RLOO regardless of the structure of $f$. In practice, this parameter can be optimized simultaneously with the model parameters (c.f. REBAR/RELAX).
>
> > Could these estimators be applied to belief nets, or would be number of discrete latents be problematic?
>
> That is an interesting future direction, and we do not suspect that the number of discrete latents would be a problem.
>
> > On strengths and weaknesses, one thing that could be discussed in more detail is the relative computational cost of all of the estimators being discussed in this paper (as there are many). For the DisARM-Tree estimator, is there any cost to calculating the tree probabilities (if C was extremely large, perhaps this becomes problematic)?
>
> Thank you for the suggestion. For the settings we explore, there is a small amount of numerical overhead with SB and Tree couplings, however, it is dwarfed by the (relatively) expensive NN evaluations, resulting in similar computation time for all methods. Computing the tree probabilities takes linear time in C, similar to computing the softmax probabilities needed to sample from the categorical variables. So while extremely large C could become problematic, that would also affect RLOO. We will add this discussion to the next revision.

---

### Author Response · Authors · 2021-08-11
**Limitations**

As requested by several of the reviewers, we provide a discussion of the limitations of our methods here and will include it in the next revision of the manuscript.

Parameterizing Categorical variables in terms of Bernoulli variables requires imposing structure on the Categorical space (e.g. an ordering or a tree), which is not fully satisfactory because in most settings there is no such natural structure for Categorical spaces. Developing coupling-based estimators that do not rely on such a structure would be interesting future work which might lead to further improvements.

While we see that our coupling-based estimators generally outperform or perform at least as well as RLOO in our experiments, using coupled samples instead of independent samples is not guaranteed to lead to better performance in every case. Learning the couplings would provide a way of ensuring an improvement over RLOO.

Finally, the estimators we propose in this paper, like all multisample estimators, can be used for RL only if the environment is simulated or we have a model of it, as they require being able to perform multiple rollouts from the same state.

---

### Decision · Program_Chairs · 2021-09-28

**Decision:**

Accept (Poster)

**Comment:**

The submission is concerned with low-variance gradient estimators for discrete random variables, extending recent work on binary to categorical. The reviewers agreed the work here is novel and interesting, but shared concerns around the lack of baselines and clarity.  The authors promised to improve clarity, and add more experimental results [but did not post these in the rebuttal]. An additional expert review was sought and the AC agrees with the expert reviewer that the submission’s overall contribution is enough to justify acceptance. The authors are strongly encouraged to improve the writing and add more experiments, as promised to the reviewers.

**Consistency Experiment:**

NeurIPS has a long history of experimentation. In 2014, NeurIPS ran an experiment in which 10% of submissions were reviewed by two independent committees to quantify the randomness in the review process. This year, we repeated a variant of this experiment to see how the quality of the review process has changed over time.  This paper was part of the experiment and was therefore assigned to two committees (consisting of reviewers, an Area Chair, and a Senior Area Chair) that reached independent decisions.  If both committees made the same recommendation, this recommendation was followed. If a single committee recommended acceptance, the paper was accepted (with the exception of a few cases in which the other committee identified what we considered a fatal flaw, e.g., an error in a key result).

Both committees reached the same decision: **Accept (Poster)**

The other committee assigned to the paper recommended **Accept (Poster)**.  You can find the other set of reviews, along with any follow up discussion with the authors here:
https://openreview.net/forum?id=byizK1OI4xA